# Divergent mechanisms regulate conserved cardiopharyngeal development and gene expression in distantly related ascidians

Alberto Stolfi[1†], Elijah K Lowe[2†], Claudia Racioppi[3], Filomena Ristoratore[3], C Titus Brown[2,4]*, Billie J Swalla[5,6]*, Lionel Christiaen[1]*

[1]Center for Developmental Genetics, Department of Biology, New York University, New York, United States; [2]Department of Computer Science and Engineering, Michigan State University, East Lansing, United States; [3]Cellular and Developmental Biology Laboratory, Stazione Zoologica Anton Dohrn, Napoli, Italy; [4]Department of Microbiology and Molecular Genetics, Michigan State University, East Lansing, United States; [5]Department of Biology, University of Washington, Seattle, United States; [6]Friday Harbor Laboratories, University of Washington, Friday Harbor, United States

**Abstract** Ascidians present a striking dichotomy between conserved phenotypes and divergent genomes: embryonic cell lineages and gene expression patterns are conserved between distantly related species. Much research has focused on Ciona or Halocynthia spp. but development in other ascidians remains poorly characterized. In this study, we surveyed the multipotent myogenic B7.5 lineage in Molgula spp. Comparisons to the homologous lineage in Ciona revealed identical cell division and fate specification events that result in segregation of larval, cardiac, and pharyngeal muscle progenitors. Moreover, the expression patterns of key regulators are conserved, but cross-species transgenic assays uncovered incompatibility, or 'unintelligibility', of orthologous cis-regulatory sequences between Molgula and Ciona. These sequences drive identical expression patterns that are not recapitulated in cross-species assays. We show that this unintelligibility is likely due to changes in both cis- and trans-acting elements, hinting at widespread and frequent turnover of regulatory mechanisms underlying otherwise conserved aspects of ascidian embryogenesis.

**\*For correspondence:**
ctb@msu.edu (CTB);
bjswalla@u.washington.edu
(BJS); lc121@nyu.edu (LC)

†These authors contributed equally to this work

**Reviewing editor**: Margaret Buckingham, Institut Pasteur, France

## Introduction

Transcriptional regulatory networks are indispensable for the generation of the diverse germ layers, anatomical structures, and cell types of multicellular organisms (*Levine and Davidson, 2005*). The impact of *cis*-regulatory DNA sequence changes on the evolution of development is undeniable but not yet fully understood (*Wray, 2007*; *Wittkopp and Kalay 2011*). Phenotypic differences between species have often been traced to differences in expression of orthologous genes. In turn, this differential expression can be attributed to changes in *cis*-regulatory modules (or 'enhancers'), which may alter their binding by sequence-specific transcription factors (TFs) (*Sucena and Stern, 2000*; *Gompel et al., 2005*; *Prud'homme et al., 2006*; *Miller et al., 2007*; *Shirangi et al., 2009*).

In contrast, gene expression patterns may be identical *in spite* of differences in the *cis*-regulatory DNA sequences controlling their transcription. For instance, orthologous enhancers might be interchangeable between different species without conspicuous DNA sequence similarity (*Maduro and Pilgrim, 1996*; *Ludwig et al., 1998*; *Piano et al., 1999*; *Romano and Wray, 2003*; *Oda-Ishii et al., 2005*). This is often attributed to the conservation of gene regulatory networks (GRNs) and the flexibility of TF binding site distribution in a given enhancer, which contribute to conservation of enhancer function (*Ludwig et al., 2000*; *Oda-Ishii et al., 2005*; *Hare et al., 2008*; *Weirauch and Hughes, 2010*).

**eLife digest** When two species have features that look similar, this may be because the features arise by the same processes during development. Other features may look similar yet develop by different mechanisms. 'Developmental system drift' refers to the process where a physical feature remains unaltered during evolution, but the underlying pathway that controls its development is changed. However, to date, there have been only a few experimental studies that support this idea.

Ascidians—also commonly known as sea squirts—are vase-like marine creatures, which start off as tadpole-like larvae that swim around until they find a place to settle down and attach themselves. Once attached, the sea squirts lose the ability to swim and start feeding, typically by filtering material out of the seawater. Sea squirts and their close relatives are the invertebrates (animals without backbones) that are most closely related to all vertebrates (animals with backbones), including humans. Furthermore, although different species of sea squirt have almost identical embryos, their genomes are very different.

Stolfi et al. have now studied whether developmental system drift may have occurred during the evolution of ascidians, by analyzing different species of sea squirt named Molgula and Ciona. Stolfi et al. compared the genomes of Molgula and Ciona and studied the expression of genes in the cells that give rise to the heart and the muscles of the head. As an embryo develops, specific genes are switched on or off, and these patterns of gene activation were broadly identical in the two species of sea squirt examined.

Enhancers are sequences of DNA that control when and how a gene is switched on. Given the similarities between the development of heart and head muscle cells in the different sea squirts, Stolfi et al. looked to see if the mechanisms of gene expression, and therefore the enhancers, were also conserved. Unexpectedly, this was not the case. When enhancers from Molgula were introduced into Ciona (and vice versa), these sequences were unable to switch on gene expression—thus enhancers from one sea squirt species could not function in the other.

Stolfi et al. conclude that the developmental systems may have drifted considerably during evolution of the sea squirts, in spite of their nearly identical embryos. This reinforces the view that different paths can lead to the formation of similar physical features.

This cryptic regulatory turnover is termed 'developmental system/systems drift' (DSD) (*True and Haag, 2001*). This term broadly applies to the divergence in the molecular or morphogenetic basis for the development of identical homologous characters.

However, gene expression patterns that appear conserved between species may also be the output of divergent regulatory networks (*Ludwig et al., 2005*). In such cases, one species' *trans* environment may be unable to interpret the other species' enhancer, even if the expression patterns controlled by the orthologous enhancers are identical between the two species. This particularly acute manifestation of DSD is thought to contribute to developmental defects in interspecies hybrids (*Takano, 1998*), contributing to reproductive isolation and speciation as *cis/trans* combinations become so incompatible as to be lethal (*Porter and Johnson, 2002*). There is ample evidence that DSD is pervasive in metazoan evolution (*Kiontke et al., 2007*; *Verster et al., 2014*), but it has been difficult to study due to its cryptic nature.

Sea squirts, or ascidians, are well suited for investigating the evolution of gene regulation in development (*Satoh, 2013*). Species on the opposite branches of the ascidian tree, estimated to have diverged ~520 million years apart, have extremely conserved embryos that share identical stereotyped cell divisions up until the late gastrula stage and beyond (*Swalla, 2006*; *Lemaire, 2009*). This allows one to unequivocally infer homology between specific embryonic cells, gene expression patterns, and GRNs. This is in contrast to the striking lack of non-coding sequence conservation, revealed by bioinformatic alignment of genomic sequences from disparate ascidian species. This means that any two randomly selected ascidian species are likely to be more genetically divergent from one another than humans are from fish, which share several highly conserved non-coding sequences (*Woolfe et al., 2004*). This dichotomy between near-identical embryos and un-alignable *cis*-regulatory sequences represents a singular paradox in the study of evolution and developmental biology today (*Lemaire, 2011*).

Modern molecular tools for embryo manipulation and visualization are available for several ascidian species, including a protocol for transfection *en masse* by electroporation of plasmid DNA into fertilized eggs of the solitary ascidian *Ciona intestinalis* (*Corbo et al., 1997*; *Christiaen et al., 2009b*). This experimental tractability has allowed researchers to begin addressing the ascidian embryological paradox. Regulation of the *Otx* gene is a prime example, as enhancers upstream of the *Otx* genes from *C. intestinalis* and the distantly related *Halocynthia roretzi* do not show any DNA sequence similarity but are broadly functional in cross-species assays. This was shown to be due to conservation of *cis*-regulatory logic in spite of a reshuffling of functional TF binding sites (*Oda-Ishii et al., 2005*). On the other hand, comparisons of *C. intestinalis* and *H. roretzi* development have also revealed clear examples of acute DSD in regulation of the *Brachyury* gene (*Takahashi et al., 1999*), cell signaling upstream of pigment cell differentiation (*Darras and Nishida, 2001*; *Abitua et al., 2012*), secondary tail muscle specification (*Kim and Nishida, 2001*; *Tokuoka et al., 2007*; *Hudson et al., 2007*; *Hudson and Yasuo, 2008*), and embryonic marginal zone patterning (*Takatori et al., 2010*; *Hudson et al., 2013*).

These few examples suggest that considerable DSD may have occurred during the evolution of ascidians, even with their presupposed highly constrained mode of embryogenesis. To get a better idea of how DSD may have shaped ascidian evolution, it was necessary to identify species comparable to *Ciona* spp. in their experimental tractability but phylogenetically distant enough as to maximize the genetic differences between them.

*Molgula* spp. belong to the order Stolidobranchia like *H. roretzi,* but produce embryos that are more comparable in size and developmental rate to those of *C. intestinalis,* a member of the order Phlebobranchia (*Figure 1A*). The genus is also remarkable for containing several species that have independently evolved anural development (*Berrill, 1931*; *Swalla and Jeffery, 1990*; *Hadfield et al., 1995*; *Jeffery et al., 1999*; *Huber et al., 2000*). These anural, or 'tail-less', *Molgula* species produce immotile larvae lacking the differentiated structures that are required for swimming (*Swalla and Jeffery, 1990*; *Kusakabe et al., 1996*), and some species even appear to bypass the larval stage altogether (*Tagawa et al., 1997*; *Maliska and Swalla, 2010*). Thus, *Molgula* species comprise a unique group in which to study body plan and life cycle evolution.

In order to realize the full potential of Molgulid ascidians as model organisms for molecular and comparative developmental genetics on par with *Ciona spp.,* we sequenced and assembled the genomes of three *Molgula* species: the tailed species *M. occidentalis* and *M. oculata,* and the tail-less *M. occulta.* Moreover, we adapted the electroporation protocol to *M. occidentalis* embryos. To illustrate the power of comparative approaches between *Molgula* and *Ciona*, we characterized the B7.5 lineage in *M. occidentalis*. In both *C. intestinalis* and *H. roretzi,* this lineage is descended from the B7.5 pair of blastomeres of the early gastrula embryo and gives rise to the anterior tail muscles of the larva, and to the heart and atrial siphon/pharyngeal muscles of the adult (*Hirano and Nishida, 1997*; *Davidson and Levine, 2003*; *Stolfi et al., 2010*). A growing body of research has focused on the molecular basis of B7.5 lineage development in *C. intestinalis* (reviewed in *Tolkin and Christiaen, 2012*; *Cota et al., 2013*), some of which have guided the discovery of vertebrate heart developmental mechanisms (*Islas et al., 2012*). Given its detailed characterization in *C. intestinalis*, we chose the B7.5 lineage for an in-depth comparative analysis. Our analysis revealed a remarkable degree of conservation of the clonal topology and gene expression patterns (together referred to as the 'ontogenetic motif') underlying ascidian cardiopharyngeal mesoderm development (*Wang et al., 2013*), but also uncovered divergent regulatory mechanisms underlying conserved gene expression profiles between *Ciona* and *Molgula*.

## Results

### De novo sequencing of *Molgula spp.* genomes

Genomes of three *Molgula* species (*M. occidentalis, M. oculata,* and *M. occulta*) were sequenced using next-generation sequencing technology and assembled. A common metric for judging the quality of a genome assembly is the contig N50 length, which is determined such that 50% of the assembly is contained in contigs of this length or greater. We used the contig N50 length to select the best assembly for each species given the varying 'k' parameter (length of k-mer overlap). A 'k' of 39 yields the best assembly for both *M. occidentalis* and *M. occulta*. The best 'k' for *M. oculata* was 61. *M. occidentalis, M. occulta,* and *M. oculata* N50 lengths were approximately 26.3 kb, 13 kb, and 34 kb, respectively (*Table 1*).

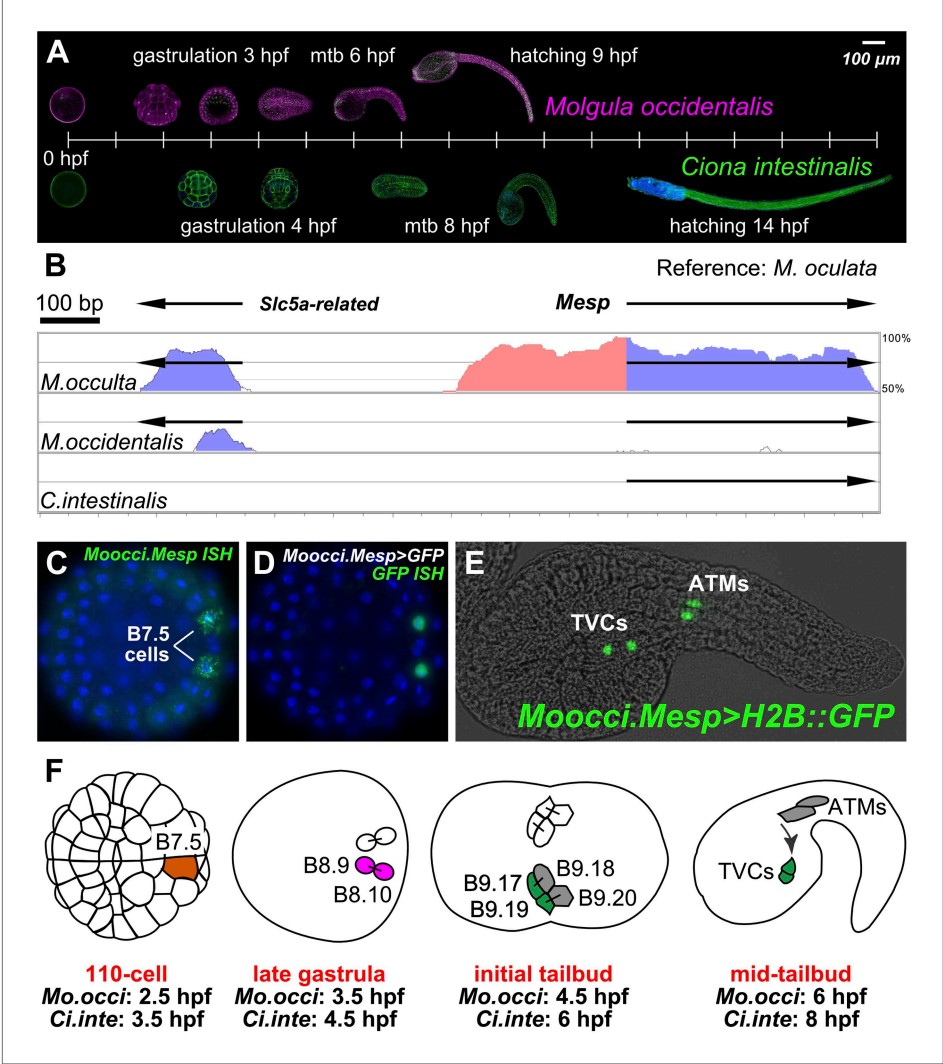

**Figure 1**. The B7.5 lineage in *M. occidentalis*. (**A**) Diagram comparing *M. occidentalis* (top) and *C. intestinalis* (bottom) embryogenesis at 24°C. Embryos were stained with Alexa Fluor dye-conjugated phalloidin to visualize cell outlines and DAPI to visualize cell nuclei. (**B**) Diagram of mVISTA (***Frazer et al., 2004***; genome.lbl.gov/vista/) alignment of *M. oculata Mesp* (*Moocul.Mesp*) locus to orthologs in *M. occulta, M. occidentalis,* and *C. intestinalis*. Shaded peaks indicate sequence conservation above 70% over 100-bp windows (blue = protein-coding, pink = non-coding). Arrows indicate direction of transcription of protein-coding genes. Non-coding sequences upstream of *Mesp* are only conserved between *M. oculata* and *M. occulta*. *M. occidentalis* and *C. intestinalis* show considerable divergence even in protein-coding sequences. Note that microsynteny with *SLC5A-related* gene supports the orthology of these sequences among the Molgulids. (**C**) In situ hybridization (ISH) for *Moocci.Mesp* in 110-cell stage embryo (vegetal view), showing mRNA detection (green) in B7.5 blastomeres. Nuclei were counterstained with DAPI (blue). Staging is given by hours post-fertilization (hpf). (**D**) Vegetal view of a 110-cell stage embryo electoporated with *Moocci.Mesp>GFP* reporter construct. Reporter gene expression was detected by ISH for *GFP* transcripts (green). Nuclei were stained with DAPI (blue). (**E**) Lateral view of a mid-tailbud stage embryo electoporated with *Moocci. Mesp>Histone2B::GFP* reporter construct. GFP fluorescence reveals B7.5 descendants on left side of embryo: two trunk ventral cells (TVCs) and two anterior tail muscles (ATMs). (**F**) Diagram of B7.5 lineage divisions from 110-cell stage to mid-tailbud stage, inferred from previous *C. intestinalis* studies. Cells are named according to Conklin's method (***Conklin, 1905***). The lineage is bilaterally symmetric, but only cells on the left side are indicated and named. Relative staging given for *M. occidentalis (Mo.occi)* and *C. intestinalis (Ci.inte)*. 110-cell and late gastrula: vegetal view. Initial tailbud: dorsal view. Mid-tailbud: lateral view. Anterior pole is on the left in all images and illustrations.

*Figure 1. Continued on next page*

*Figure 1. Continued*

The following figure supplements are available for figure 1:

**Figure supplement 1**. Alignment of 5′ flanking sequences from *Mesp* orthologs.

**Figure supplement 2**. The B7.5 lineage of the tail-less species *M. occulta*.

**Figure supplement 3**. *Moocci.Mesp* in situ hybridization at tailbud stage.

---

In addition to N50 lengths, we also used CEGMA (Core Eukaryotic Genes Mapping Approach) scores, in order to evaluate the assemblies' representative completeness (*Parra et al., 2007*). CEMGA reports scores for complete and partial alignments to a subset of core eukaryotic genes. An alignment is considered 'complete' if at least 70% of a given protein model aligns to a contig in the assembly, while a partial alignment indicates that a statistically significant portion of the protein model aligns. The partial alignment scores are ~97% or higher for all assemblies. *M. oculata* has the best complete alignment score at ~90%. *M. occidentalis* and *M. occulta* have complete alignment scores of 81% and 77% respectively (*Table 1*). These scores indicate that our assemblies contain at least partial sequences for the vast majority of protein-coding genes in the genomes of these species.

Various factors make it unreliable to predict genome size and gene density based on assembly metrics alone (*Bradnam et al., 2013*). Of the handful of sequences we isolated and analyzed, we found that the sizes of introns and upstream regulatory regions were roughly comparable to those from their *Ciona* orthologs. This suggests that the *Molgula* genomes may be as compact as the *C. intestinalis* genome (i.e., ~150–170 Mb, ~16,000 genes, *Laird, 1971*; *Simmen et al., 1998*; *Satou et al., 2008*).

Our sequencing efforts revealed extreme genetic divergence not only between *Ciona* and *Molgula,* as expected, but even within the Molgulids. For example, we used BLAST to identify the *Molgula* orthologs of *C. intestinalis Mesp* (*Ciinte.Mesp,* as per the proposed tunicate gene nomenclature rules, see *Stolfi et al., 2014*). *Ciinte.Mesp* is the sole ortholog of vertebrate genes coding for MesP and Mesogenin bHLH transcription factor family members (*Satou et al., 2004*). VISTA alignment shows high sequence similarity between sequences 5′ upstream of the *Mesp* genes from the closely related *M. oculata* and *M. occulta* (*Figure 1B*). However, there is no conservation of *Mesp* DNA sequences, coding or non-coding, between *M. oculata/occulta* and *M. occidentalis*, nor between *C. intestinalis* and any of the three *Molgula* species (*Figure 1—figure supplement 1*). In previous phylogenetic surveys, *M. occidentalis* has been placed as an early-branching *Molgula* species, often grouped together in a subfamily with species ascribed to the genera *Eugyra* and *Bostrichobranchus* instead (*Hadfield et al., 1995*; *Huber et al., 2000*; *Tsagkogeorga et al., 2009*). Our sequencing results support the view that *M. occidentalis* is highly diverged from other *Molgula* spp.

## Expression of the *M. occidentalis Mesp* gene marks the B7.5 cells

*Ciinte.Mesp* specifies the B7.5 cells as the sole progenitors of the cardiopharyngeal lineage (*Satou et al., 2004*; *Davidson et al., 2005*; *Hirano and Nishida, 1997*; *Stolfi et al., 2010*). We performed

---

**Table 1.** Genome assembly statistics

| Species | N50 | Mean contig length | Total | Total number of base pairs | CEGMA C[1] | CEGMA P[2] |
|---------|-----|--------------------|-------|-----------------------------|-----------|-----------|
| *M. occidentalis* | 26,298 | 5072 | 51,761 | 262,547,660 | 81.45 | 96.77 |
| *M. occulta* | 13,011 | 3233 | 58,489 | 189,110,562 | 77.42 | 98.79 |
| *M. oculata* | 34,042 | 6270 | 25,497 | 159,886,716 | 89.92 | 99.19 |

The contig N50 length, mean contig length, total number of contigs, total number of base pairs and CEGMA scores were collected for each draft assembly. The CEGMA scores is a metric of completeness measured against highly Conserved eukaryotic genes. Alignments of 70% or greater of the protein length are called complete (C[1]) and all other statistically significant alignments are called partial (P[2]).

RNA in situ hybridization (ISH) for *M. occidentalis Mesp (Moocci.Mesp)* and found that this gene is also expressed only in the B7.5 cells of *M. occidentalis* embryos (*Figure 1C*). These cells are unequivocally identified due to the perfect conservation of early embryonic cell cleavage patterns in all ascidians. ISH for the *M. occulta Mesp* gene *(Mooccu.Mesp)* also revealed conserved expression in this tail-less species (*Figure 1—figure supplement 2A*).

We successfully adapted the *Ciona* electroporation protocol for simultaneous transfection of reporter gene plasmids into hundreds of synchronized *M. occidentalis* embryos (*Figure 1D,E*). We were also able to electroporate *M. occulta* embryos (*Figure 1—figure supplement 2B*). However, only *M. occidentalis* was routinely available to us for in vivo studies, so we focused our experiments on this species. Development of *M. occidentalis* embryos was optimal at 24°C and faster than that of *C. intestinalis* (*Figure 1A*). Using electroporation-based transfection, we determined that an ~1.1 kb genomic DNA fragment upstream of *Moocci.Mesp* is able to drive expression of fused reporter genes specifically in the B7.5 cells with no 'leaky' expression in other cells as is commonly observed in *C. intestinalis* (*Figure 1D*; *Stolfi and Christiaen, 2012*).

This faithful recapitulation of *Moocci.Mesp* expression and the persistence of GFP allows for visualization of the descendants of B7.5 long after endogenous *Moocci.Mesp* transcription has ceased (*Figure 1—figure supplement 3*; *Davidson et al., 2005*). At the tailbud stage, we find that each B7.5 blastomere gives rise to four grand-daughter cells (*Figure 1E*). The two anterior B7.5 grand-daughter cells on either side of the bilaterally symmetric embryo migrate anteriorly and are termed the trunk ventral cells (TVCs) due to their final position in the *C. intestinalis* and *H. roretzi* embryos (*Nishida, 1987*). Their posterior sister cells remain in the tail and become anterior tail muscles (ATMs). As far as we can tell, B7.5 lineage ontogeny is perfectly conserved between *M. occidentalis* and *C. intestinalis* (*Figure 1F*).

## Cardiopharyngeal mesoderm gene expression

Our focus shifted to the TVCs, which in *H. roretzi* and *C. intestinalis* have been shown to be multipotent cardiopharyngeal progenitor cells that give rise to the heart and pharyngeal muscles of the atrial siphon (*Hirano and Nishida, 1997*; *Stolfi et al., 2010*). We performed a small ISH screen for orthologs of transcription factors that are expressed in *C. intestinalis* TVCs (*Satou et al., 2004*; *Davidson et al., 2005*; *Davidson and Levine, 2003*; *Christiaen et al., 2008*; *Beh et al., 2007*). These include the orthologs of conserved cardiac regulators *Ets1/2*, *GATA4/5/6*, and *NK4/Nkx2.5/tinman* (*Cripps and Olson, 2002*) (*Figure 2*).

In *M. occidentalis, Ets.b* expression is initiated in B7.5 blastomeres (*Figure 2—figure supplement 1*), is maintained in their daughter cells the cardiopharyngeal founders and in the TVCs during their migration (*Figure 2A*). This profile is similar to the expression of *C. intestinalis Ets.b (Ciinte.Ets.b)*, previously named *Ets/pointed2* or *Ets1/2* (see *Supplementary file 2* for list of old and new gene name correspondences). In *C. intestinalis,* Ets.b mediates the FGF/MAPK-dependent induction of TVCs in part by the activation of key regulators such as *Foxf, Hand-related (Hand-r,* also known as *Hand-like* or *NoTrlc)*, and *Gata4/5/6* (also known as *GATA-a*) prior to the onset of TVC migration (*Davidson et al., 2006*; *Beh et al., 2007*). In *M. occidentalis,* orthologs of *Foxf* and *Hand-r* are also activated in the TVCs shortly before and throughout their migration away from the ATMs (*Figure 2B,C*). *Moocci.Gata4/5/6* expression was detected in migrating TVCs but not before migration (*Figure 2D*). This is slightly different from *Ciinte.Gata4/5/6*, which is expressed in *C. intestinalis* TVCs prior to migration (*Christiaen et al., 2010*; *Ragkousi et al., 2011*). Expression of *Moocci. Foxf* and *Moocci.Gata4/5/6* in surrounding epidermis and endoderm, respectively, is identical to the expression domains of their orthologs in *C. intestinalis* and sometimes obscured TVC expression. However, double ISH/immunohistochemical detection (ISH/IHC) of *Moocci.Mesp* promoter-driven reporter gene clearly shows transcripts in the migrating B7.5-derived TVCs (*Figure 2—figure supplement 2A–C*).

*Foxf* and *Hand-r* are also expressed in the TVCs of tail-less *M. occulta* embryos. Because these embryos do not extend a tail and lack functional larval tail muscles, it is unclear whether their TVCs move away from the posterior pole of the embryo (*Figure 2—figure supplement 3A–F*). However, their TVCs occupy the ventro-lateral regions of the trunk (*Figure 1—figure supplement 2B*), similar to what is observed in *M. occidentalis,* suggesting they might be migrating. We cannot draw any further conclusions about the TVCs of *M. occulta,* but have not found any evidence for fundamental molecular or morphological differences relative to *M. occidentalis.*

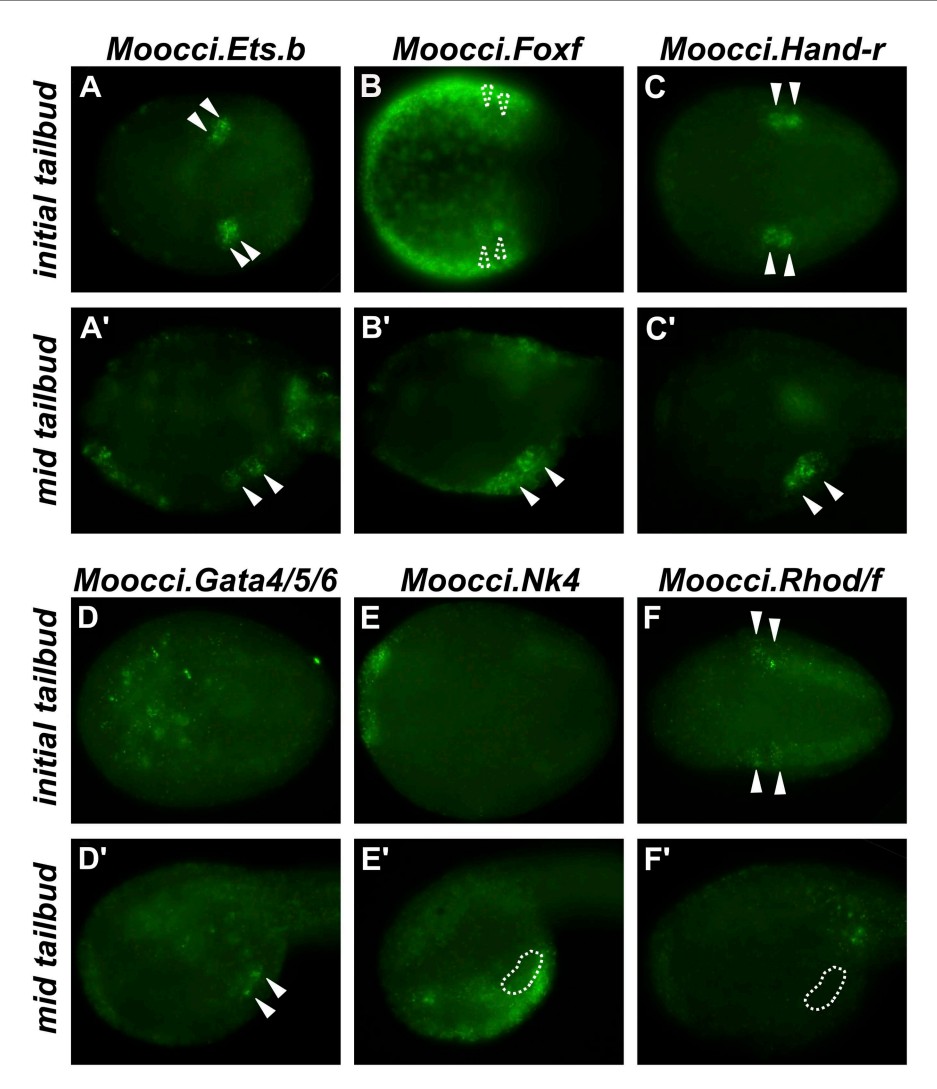

**Figure 2**. Expression of conserved TVC/heart markers in *M. occidentalis* embryos. In situ hybridization (ISH) in *M. occidentalis* embryos for (**A** and **A'**) *Moocci.Ets.b*, (**B** and **B'**) *Moocci.Foxf*, (**C** and **C'**) *Moocci.Hand-related (Moocci.Hand-r)*, (**D** and **D'**) *Moocci.Gata4/5/6*, (**E** and **E'**) *Moocci.Nk4*, (**F** and **F'**) *Moocci.Rhod/f*. ISH was performed on initial tailbud (**A**–**F**) and mid tailbud (**A'**–**F'**) stage embryos. Solid arrowheads indicate definitive expression in TVCs. Dotted arrowheads indicate potential expression of *Moocci.Foxf* in initial tailbud, obscured by strong epidermal expression. Dotted outline indicates probable position of TVCs, not visible due to lack of mRNA hybridization signal. Initial tailbud embryos were imaged ventrally or dorsally, while tailbud embryos were imaged laterally.

The following figure supplements are available for figure 2:

**Figure supplement 1**. *Ets.b* expression in B7.5 of *M. occidentalis*.

**Figure supplement 2**. In situ hybridizations reveal TVC gene expression in *M. occidentalis*.

**Figure supplement 3**. *Hand-r* and *FoxF* co-expression reveals TVCs of *M. occulta*.

**Figure supplement 4**. *Aldh1a* expression in *M. occidentalis* embryo.

Notably, the ATMs appear to be specified in *M. occulta,* as revealed by expression of *Aldh1a* (also known as *Raldh2*, **Figure 2—figure supplement 3G–I**), which encodes the rate-limiting enzyme for retinoic acid (RA) synthesis. Therefore, although the larval muscles of *M. occulta* do not differentiate

(*Swalla and Jeffery, 1990*; *Kusakabe et al., 1996*), their ATMs may still be required for RA-mediated embryonic patterning like in *C. intestinalis* (*Nagatomo and Fujiwara, 2003*) and probably also in *M. occidentalis* (*Figure 2—figure supplement 4*).

Unexpectedly, we could not detect unequivocal expression of the sole *M. occidentalis* ortholog of *NK4/Nkx2.5/tinman* (*Moocci.Nk4*) in the TVCs (*Figure 2E*). Expression in the ventral ectoderm mirrored that seen in *C. intestinalis*, indicating that the probe used was functional. It is possible that expression of *Nk4* in the TVCs of *M. occidentalis* is below the threshold of reliable detection by ISH. Our ISH data are not entirely incompatible with *Moocci.Nk4* being expressed in the TVCs at very low levels (*Figure 2—figure supplement 2D*).

Conserved TVC gene expression patterns were not limited to TF-coding genes. The small GTPase-coding *Rhod/f* gene was found to be transcriptionally upregulated in *M. occidentalis* TVCs just prior to migration (*Figure 2F*). In *C. intestinalis,* Rhod/f was identified as an effector of cytoskeleton dynamics in TVC migration and a direct transcriptional target of Foxf and activated Ets.b (*Christiaen et al., 2008*). This suggests that the conservation of B7.5 development in ascidians extends to the interface between transcriptional regulators and cellular effectors of cell migration.

## Specification of heart and pharyngeal muscles from common progenitors

In *C. intestinalis,* heart and atrial siphon (pharyngeal) muscle precursors are derived from TVCs in a two-step process involving asymmetric cell divisions (*Stolfi et al., 2010*; *Wang et al., 2013*). First, each TVC undergoes an asymmetric cell division along the medial/lateral (M/L) axis to produce a medial First Heart Precursor (FHP) and a lateral Secondary TVC (STVC). Each STVC then also undergoes a M/L asymmetric division to give rise to a medial Second Heart Precursor (SHP) and a lateral atrial siphon muscle founder cell (ASMF). In *C. intestinalis,* this step-wise segregation of heart vs pharyngeal muscle fate is coordinated by (1) activation of *Tbx1/10* in the STVCs (*Wang et al., 2013*) and (2) Tbx1/10-dependent activation of *Collier/Olf/EBF (Ciinte.Ebf,* previously named *COE*) in the ASMFs (*Stolfi et al., 2010*; *Wang et al., 2013*; *Razy-Krajka et al., 2014*). The shared clonality of heart and pharyngeal muscles and similarity of associated TF expression patterns point to a single evolutionary origin for this cardiopharyngeal 'ontogenetic motif' in the last common ancestor of tunicates and vertebrates (*Wang et al., 2013*).

In *M. occidentalis,* we found that the segregation of heart and pharyngeal muscle fates occurs in identical fashion to that of *C. intestinalis,* albeit on a faster timescale. At 6 hr post-fertilization (hpf) the first asymmetric division of the TVCs begins, with each TVC giving rise to a smaller, more ventral/medial cell and a larger, more dorsal/lateral cell. At 7.5 hpf, the resulting STVCs also undergo an asymmetric division along the same axis (*Figure 3A*). The end result is a cluster consisting of two larger cells (presumptive ASMFs) lateral to four smaller cells (presumptive heart precursors), on either side of the embryo (*Figure 3B*).

Gene expression during heart vs pharyngeal muscle segregation is also conserved. Namely, *Moocci. Tbx1/10* is specifically activated in the STVCs immediately following the first asymmetric division (*Figure 3C*), while *Moocci.Ebf* is specifically activated in the ASMFs immediately following the second asymmetric division (*Figure 3D*). *Moocci.Ebf* expression is maintained in ASMFs and their daughter cells, the atrial siphon muscle precursors (ASMPs) during their migration to the dorsal regions of the now swimming larva (*Figure 3E*). This step-wise progression of gene expression, cell division, and migration are virtually identical to those observed in *C. intestinalis,* demonstrating deep evolutionary conservation of the cardiopharyngeal ontogenetic motif in ascidians.

We allowed embryos electroporated with *Moocci.Mesp>H2B::GFP* to settle and undergo metamorphosis from the larval to the juvenile stage. Visualization of H2B::GFP revealed contributions of the B7.5 lineage to the heart and pharyngeal muscles of juveniles (*Figure 3F*). Thus, the post-metamorphic tissues derived from B7.5 are identical to those previously reported for *C. intestinalis* and *H. roretzi* (*Hirano and Nishida, 1997*; *Stolfi et al., 2010*). A summary of the TVC divisions and their post-metamorphic fates is shown in *Figure 3G*. Based on the perfectly conserved parallels to *C. intestinalis*, we hypothesize that specification of ASM vs heart fate in *Molgula* occurs through a conserved Tbx1/10-Ebf-dependent regulatory cascade.

In summary, the cardiopharyngeal mesoderm of *Molgula* and *Ciona* display unequivocally homologous ontogenies, as revealed by nearly identical cell divisions, cell fate choices, and gene expression patterns. These data indicate that the key developmental features of the B7.5 lineage evolved at the base of the tunicate radiation.

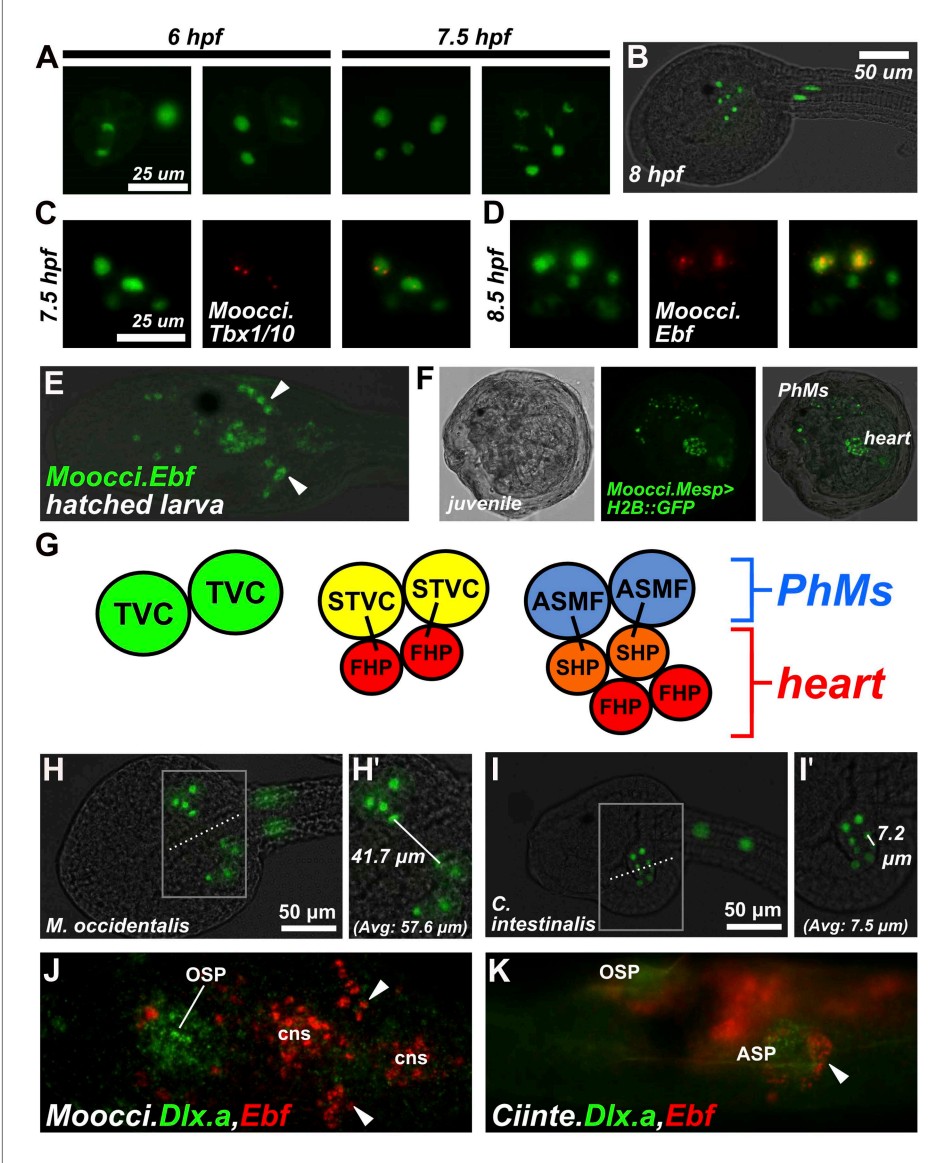

**Figure 3**. Specification of heart vs atrial siphon muscle precursors in *M. occidentalis*. (**A**) First and second asymmetric division of TVCs in *M. occidentalis* embryos. B7.5 lineage was visualized by electroporation of *Moocci.Mesp>H2B::GFP* (green). The first division occurs at ~6 hpf at 24°C, and the second division occurs at ~7.5 hpf at 24°C. (**B**) Result of two asymmetric divisions of TVCs on left side of embryo electroporated with *Moocci.Mesp>H2B::GFP* (green). At 8 hpf at 24°C, a cluster of 6 cells derived from the TVCs is located in the ventro-lateral region of the trunk. From top to bottom: 2 atrial siphon muscle founder cells (ASMFs), 2 second heart precursors (SHPs), and 2 first heart precursors (FHPs). (**C**) In situ hybridization (ISH, red) for *Moocci.Tbx1/10* + β-galactosidase immunodetection (IHC, green) in embryos electoporated with *Moocci.Mesp>nls::lacZ*. *Moocci.Tbx1/10* nascent transcripts are detected as two dots in the nuclei of Secondary TVCs (STVCs), between the first and second asymmetric divisions. (**D**) ISH + IHC for *Moocci.Ebf* (red) in embryos electoporated with *Moocci.Mesp>nls::lacZ* (green), revealing *Moocci.EBF* expression in ASMFs after the second asymmetric division. (**E**) ISH for *Moocci.Ebf* in a swimming larva, viewed dorsally, revealing *Moocci.Ebf+* migrating atrial siphon muscle precursors (ASMPs, arrowheads). (**F**) Lateral view of a *M. occidentalis* juvenile (>100 hpf) electroporated with *Moocci.Mesp>H2B::GFP*. GFP + nuclei reveal contributions of B7.5 lineage to atrial siphon muscle-derived pharyngeal muscles (PhMs) and heart. (**G**) Diagram of the TVC divisions giving rise to the pharyngeal muscles and heart of the adult. (**H**) Ventral view of a *M. occidentalis* embryo electroporated with *Moocci.Mesp>H2B::GFP*, just after the first asymmetric division of the TVCs. (**H'**) Inset from (**H**) is focused on the distance between FHPs on either side of the embryo (mean = 57.6 μm, n = 9). (**I**) Ventrolateral view of a *C. intestinalis* embryo electroporated with *Ciinte.Mesp>H2B::GFP*, *Figure 3. Continued on next page*

*Figure 3. Continued*

right after the first asymmetric division. (**I'**) The distance between FHPs from either side is very small (mean = 7.5 μm, n = 14) since they contact each other at the midline to form a single cluster of cells. (**J**) Double ISH for *Moocci.Dlx.a* (green) and *Moocci.Ebf* (red) in a *M. occidentalis* larva, viewed dorsally. *Moocci.Dlx.a* marks the siphon primordia, while *Moocci.Ebf* marks siphon muscle precursors. Arrowheads point to atrial siphon muscle precursors (ASMPs), which have migrated dorsally but do not encounter an atrial siphon primordium around which to encircle. In contrast, oral siphon muscle precursors form a ring around the oral siphon primordium (OSP). Other *Moocci.Ebf+* cells seen are neurons or neuronal precursors in the central nervous system (cns). (**K**) Double ISH for *Ciinte.Dlx.a* (green) and *Ciinte.Ebf* (red) in a *C. intestinalis* larva viewed dorsolaterally. Arrowhead indicates *Ciinte.Ebf+* ASMPs on one side of the embryo encircling one of two bilaterally paired atrial siphon primordia (ASP). The same configuration of *Ciinte.Ebf+* muscle precursors is seen encircling the OSP.

The following figure supplements are available for figure 3:

**Figure supplement 1**. Mosaic transgene labeling of *M. occidentalis* suggests bilateral origin of juvenile heart.

**Figure supplement 2**. Delayed atrial siphon primordium specification in *M. occidentalis*.

**Figure supplement 3**. Cardiopharyngeal development in *C. intestinalis* vs. *M. occidentalis*.

## Species-specific differences in TVC migration and heart primordium formation

Despite perfect conservation of B7.5 lineage cell divisions and cell fates between *M. occidentalis* and *C. intestinalis*, we noticed a striking species-specific difference in TVC behavior (summarized in ***Figure 3***, ***Figure 3—figure supplement 3***). Namely, in *M. occidentalis,* the TVCs appear to migrate along a more lateral path than their counterparts in *C. intestinalis*. As a result, the TVCs on one side of the embryo do not contact those on the other side, resulting in two discrete heart precursor clusters during the tailbud stage (***Figure 3H***). In *C. intestinalis*, the TVCs converge towards the ventral midline prior to dividing, such that the resulting FHPs and SHPs from the two sides contact each other as soon as they are born, forming a single cluster of heart precursors (***Figure 3I***; ***Davidson and Levine, 2003***). However, observation of juveniles electroporated with *Mocci.Mesp>H2B::GFP* showed some individuals with hearts that were only half-labeled with H2B::GFP, likely due to left/right transgene mosaicism (***Figure 3—figure supplement 1***). This indicates that heart precursors from both sides of the embryo ultimately contribute to a single heart, as they do in *C. intestinalis.* Therefore, we assume this naturally occurring *cardia bifida* is only a temporary configuration of a single embryonic heart primordium. This odd configuration has not been previously described in the embryos of other ascidians and may represent a *Molgula*-specific feature.

## Differences in ASMP behavior related to heterochrony of atrial siphon formation

We also noticed a difference in ASM precursor behavior. In *C. intestinalis*, migrating ASMPs from either side of the embryo migrate dorsally, forming rings around a bilateral pair of atrial siphon primordia (ASPs) on the corresponding side (***Stolfi et al., 2010***). The ASPs are placode-like rosettes of molecularly and morphologically distinct ectodermal cells that arise through a retinoic acid/Hox1- and FGF/MAPK-dependent program and are proposed to be homologous to vertebrate otic placodes (***Wada et al., 1998***; ***Mazet et al., 2005***; ***Kourakis and Smith, 2007***; ***Sasakura et al., 2012***). In juveniles of *C. intestinalis* and other phlebobranch ascidians, there are initially two atrial siphons, one on either side of the animal, that eventually fuse into a single one at the end of the '1st Ascidian' stage (***Chiba et al., 2004***; ***Kourakis et al., 2010***). In contrast, all stolidobranch ascidians initially specify a single ASP, bypassing the two-siphon fusion process (***Manni et al., 2004***; ***Grave, 1926***; ***Figure 3—figure supplement 2A–C***). Indeed, we observed the formation of a single ASP in *M. occidentalis* late in metamorphosis and in the larvae of *M. oculata* (***Figure 3—figure supplement 2A,B***). This flexibility in atrial siphon development in ascidians is an example of DSD at the morphogenetic level. It is believed that the dual primordium condition is ancestral and that the specification of a single primordium is a derived, Stolidobranchia-specific trait (***Kourakis et al., 2010***).

We found that *M. occidentalis* ASMPs, marked by *Ebf* expression, migrate dorsally but remain as a cluster, unlike the characteristic rings of cells stretched around the invaginations of the ASPs, as observed in *C. intestinalis* (*Figure 3J,K*). On the other hand, oral siphon muscle precursors form a ring of *Moocci.Ebf+* cells around the presumptive oral siphon primordium (*Figure 3J*). ISH for the siphon primordium marker *Dlx.a* reveals that only the oral siphon primordium is specified at this stage in *M. occidentalis* larvae. In *C. intestinalis*, both oral and atrial siphon primordia expressing *Dlx.a* are specified by the larval stage and are encircled by *Ebf+* siphon muscle precursors (*Figure 3K*). Thus, we conclude that this difference in ASMP ring formation is related to the heterochrony of atrial siphon formation between *M. occidentalis* and *C. intestinalis*.

## Divergence of *Mesp cis*-regulatory sequence function between *M. occidentalis* and *C. intestinalis*

Given the obvious parallels between *C. intestinalis* and *M. occidentalis* cardiopharyngeal development, we expected transcriptional regulatory mechanisms to also be highly conserved between the two species. We tested this assumption by electroporating *C. intestinalis* reporter constructs into *M. occidentalis* embryos, and vice-versa. We observed that a *Ciinte.Mesp* reporter construct (*Davidson et al., 2005*), when electroporated into *M. occidentalis* embryos, drives relatively weak reporter gene expression in B7.5 with substantial leaky expression in other tissues (*Figure 4A*, *Figure 4—figure supplement 1*). Conversely, the *Moocci.Mesp* enhancer fails to drive any reporter gene expression when electroporated into *C. intestinalis* embryos (*Figure 4B*), despite recapitulating robust B7.5-specific expression in *M. occidentalis* embryos (*Figure 1D,E*).

These data suggest acute DSD of transcriptional regulatory mechanisms underlying otherwise identical *Mesp* expression patterns. More specifically, the *trans*-regulatory environment of the B7.5 blastomeres has diverged between *Molgula* and *Ciona*, and compensatory changes in the respective *Mesp cis*-regulatory sequences must have rendered these unable to function adequately outside of that milieu.

## *Tbx6-related* and *Lhx3/4* gene expression patterns in *M. occidentalis*

The drift of *Mesp* regulation between *M. occidentalis* and *C. intestinalis* prompted us to investigate potential species-specific differences in upstream *trans*-acting regulators. In *C. intestinalis*, *Mesp* is transcriptionally activated downstream of two zygotically expressed TFs: Tbx6-related.b (Ciinte.Tbx6-r.b) and Lhx3/4 (Ciinte.Lhx3/4) (*Satou et al., 2004*; *Davidson et al., 2005*; *Christiaen et al. 2009a*). *Ciinte.Tbx6-r.b* is directly activated in the posterior mesodermal lineages by the maternal determinant Macho-1/Zic-related.a (Ciinte.Zic-r.a) (*Yagi et al., 2004*), while *Ciinte.Lhx3/4* is activated in the vegetal pole in response to stabilized maternal Beta-catenin (*Satou et al., 2001*). Since the expression domains of Ciinte.Tbx6-r.b and Ciinte.Lhx3/4 overlap only in B7.5, activation of *Ciinte.Mesp* is achieved through the synergistic interaction between Ciinte.Tbx6-r.b and Ciinte.Lhx3/4 proteins at the *Ciinte.Mesp* promoter in these cells (*Christiaen et al., 2009a*).

In each of the three *Molgula* genome assemblies, we identified two *Tbx6-r* genes of uncertain orthology relationships to the four *Tbx6-r* genes in the *C. intestinalis* genome (*Stolfi et al., 2014*). In all three *Molgula* genomes, the two *Tbx6-r* genes are arranged in head-to-head configuration with ~2 kb separating the transcript start sites (*Figure 4—figure supplement 2B*). Despite the uncertain phylogeny of tunicate *Tbx6-r* genes, we named these genes *Tbx6-r.a* and *Tbx6-r.b* in *Molgula*. In *M. occidentalis*, both *Tbx6-r.a* and *Tbx6-r.b* are expressed in a broad posterior swath of the pregastrula embryo in a manner similar to the expression pattern of *Tbx6-r* genes in *C. intestinalis* (*Figure 4C,D*; *Yagi et al., 2004*).

We also identified two *Lhx3/4* genes in each *Molgula* genome. This was unexpected because no *Lhx3/4* duplications have been identified in any ascidian species (*Christiaen et al., 2009a*; *Kobayashi et al., 2010*). We named these genes *Lhx3/4.a* and *Lhx3/4.b*. In all three *Molgula* species, *Lhx3/4.a* is the more conserved paralog, while *Lhx3/4.b* is more divergent, lacking a well-conserved C-terminal motif (*Figure 4—figure supplement 2A*). However, the amino acid sequence changes are not predicted to change the DNA-binding preference of the homeodomain (*Figure 4—figure supplement 3*; *Noyes et al., 2008*). In each of the three *Molgula* species' genome assemblies, the two *Lhx3/4* genes were found to be located on separate contigs.

In *M. occidentalis*, *Lhx3/4.b* is expressed in vegetal cells of the early embryo, overlapping with *Tbx6-r* genes only in the B7.5 cells (*Figure 4F,G*). In contrast, *Moocci.Lhx3/4.a* is not zygotically

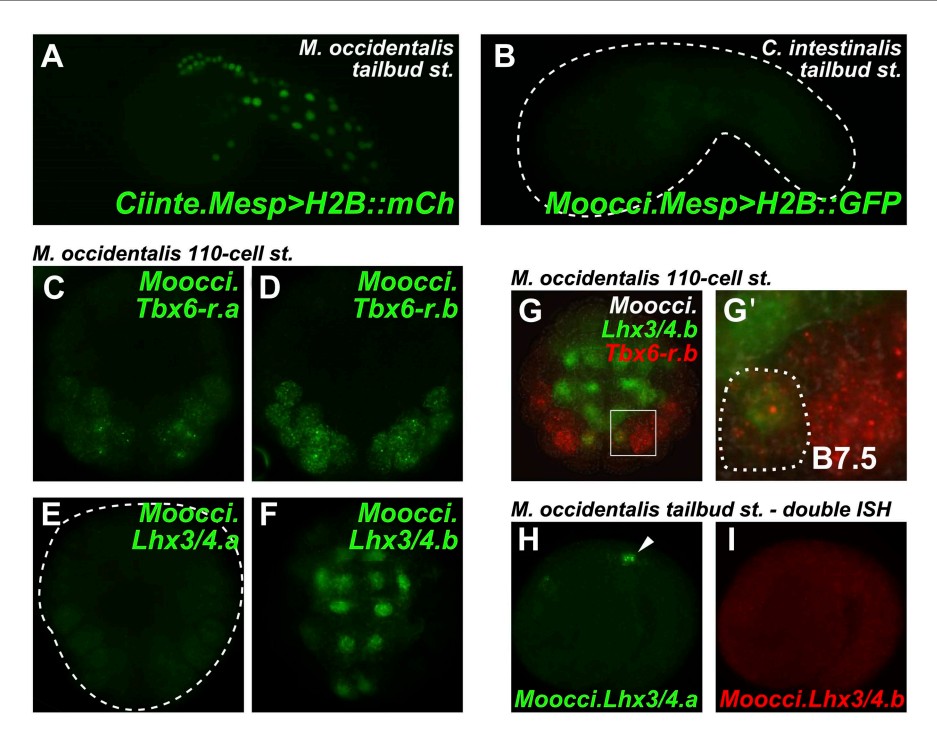

**Figure 4**. Developmental system drift of *Mesp* regulation between *C. intestinalis* and *M. occidentalis*.
(**A**) *M. occidentalis* tailbud embryo electroporated with a *Ciinte.Mesp>H2B::mCherry* reporter construct. Weak
reporter gene expression was observed in the B7.5 lineage and occasionally in other territories including
B-line mesenchyme and tail muscle cells, and A-line neural plate derivatives. (**B**) *C. intestinalis* tailbud embryo
electroporated with *Moocci.Mesp>H2B::GFP* reporter. No fluorescence was seen in any cells, indicating complete
lack of activity of *Moocci.Mesp* enhancer in wild-type *C. intestinalis* embryos. (**C**) In situ hybridization (ISH) for
*Moocci.Tbx6-r.a*, (**D**) *Moocci.Tbx6-r.b*, (**E**) *Moocci.Lhx3/4.a*, and (**F**) *Moocci.Lhx3/4.b* in 110-cell stage embryos.
(**G**) Double ISH in 110-cell stage embryo reveals co-expression of *Moocci.Lhx3/4b* (green) and *Moocci.Tbx6-r.b*
(red) exactly in the B7.5 cells of *M. occidentalis*. (**G'**) Magnified view of inset in (**G**). (**H**) Double ISH for *Moocci.
Lhx3/4.a* (green) and (**I**) *Moocci.Lhx3/4.b* (red) in a mid-tailbud embryo. *Moocci.Lhx3/4.a* but not *Moocci.Lhx3/4.b*
is expressed in motor ganglion neurons (arrowhead).

The following figure supplements are available for figure 4:

**Figure supplement 1**. Weak and leaky expression of *Ciinte.Mesp* reporter in *M. occidentalis* embryos.

**Figure supplement 2**. Configuration of Lhx3/4 protein domains and *Tbx6-r* locus in *M. occidentalis*.

**Figure supplement 3**. Divergent *Molgula* Lhx3/4.b homeodomains are not predicted to have altered DNA binding
specificities.

expressed in the early embryo, but is expressed in larval motor neurons (*Figure 4E,H*), where *Moocci.
Lhx3/4.b* is not transcriptionally active (*Figure 4I*). In both *C. intestinalis* and *H. roretzi*, transcription of
a single *Lhx3/4* gene from separate proximal and distal promoters results in two distinct transcript vari-
ants. Early vegetal cells express one transcript variant while motor neurons express the other transcript
variant (*Wada et al., 1995*; *Satou et al., 2001*; *Christiaen et al., 2009a*; *Kobayashi et al., 2010*).
Since this pleiotropic expression profile appears to be strictly partitioned between the two *Lhx3/4*
paralogs in *M. occidentalis*, this constitutes a clear example of regulatory sub-functionalization fol-
lowing gene duplication (*Ohno, 1970*; *Hahn, 2009*).

In summary, *M. occidentalis* embryos express *Tbx6-r* and *Lhx3/4* genes in early B7.5 blastomeres,
where they could potentially synergize to activate *Moocci.Mesp* following a *trans*-acting logic shared
with *C. intestinalis*. The next set of experiments sought to test this potentially conserved logic.

## Moocci.Tbx6-r.b and Moocci.Lhx3/4 proteins are conserved enough to activate *Ciinte.Mesp*

We asked whether *cis/trans* co-evolution between *Mesp* and Tbx6-r and Lhx3/4 could account for incompatibility of *Mesp* regulatory sequences between *M. occidentalis* and *C. intestinalis*. We previously demonstrated that in *C. intestinalis*, overexpression of Tbx6-r.b in the vegetal pole cells of the embryo, using the *cis*-regulatory sequences from the *Foxd.b* gene (*Shi and Levine, 2008*), results in ectopic *Mesp* activation in these cells due to expanded overlap of Tbx6-r.b and Lhx3/4 (*Christiaen et al., 2009a*). Similarly, overexpression of Lhx3/4 in the posterior pole of the embryo, using the *cis*-regulatory sequences from *Tbx6-r.b*, is sufficient to cause ectopic *Mesp* expression throughout this territory. We used these gain-of-function assays in *C. intestinalis* with the orthologous *M. occidentalis* proteins to test potential conservation of function in synergistic activation of *Mesp* expression.

Overexpression of Moocci.Tbx6-r.b but not Moocci.Tbx6-r.a was sufficient to activate ectopic *Ciinte.Mesp* reporter construct expression in vegetal pole cells (*Figure 5A–C*). In this regard, Moocci.Tbx6-r.b function seemed comparable to that of Ciinte.Tbx6-r.b (*Figure 5D*). Conversely, overexpression of either Lhx3/4.b or Lhx3/4.a from *M. occidentalis* in the posterior *C. intestinalis* embryo using the *Ciinte.Tbx6-r.b* promoter caused ectopic expression of both endogenous *Ciinte.Mesp* (*Figure 5—figure supplement 1*) and *Ciinte.Mesp* reporter construct (*Figure 5E–G*). These experiments revealed that both *M. occidentalis* Lhx3/4 proteins are conserved enough to activate low levels of *Ciinte.Mesp* expression (presumably by synergizing with endogenous Ciinte.Tbx6-r.b). However, neither was as effective as Ciinte.Lhx3/4 in activating *Ciinte.Mesp* (*Figure 5H*). These data suggested that *cis/trans* or *trans/trans* co-evolution involving the highly divergent Moocci.Lhx3/4.b paralog may only partially account for the incompatibility of *Mesp cis*-regulatory logic between the two species.

## Moocci.Tbx6-r.b but not Moocci.Lhx3/4 can activate *Moocci.Mesp* reporter expression in *C. intestinalis* embryos

Having established that *M. occidentalis* proteins can activate ectopic *Ciinte.Mesp* expression, we next asked whether these could be sufficient to activate *Moocci.Mesp* reporter expression in *C. intestinalis* embryos. Indeed, *Moocci.Mesp* reporter activation in the vegetal pole was observed upon overexpression of Moocci.Tbx6-r.b (*Figure 5J*), but not Moocci.Tbx6-r.a (*Figure 5I*), consistent with their effects on *Ciinte.Mesp* activation. Surprisingly, overexpression of Ciinte.Tbx6-r.b was not sufficient to activate *Moocci.Mesp* reporter (*Figure 5K*). These results hint at *cis/trans* co-evolution between *Moocci.Mesp cis*-regulatory sequences and Moocci.Tbx6-r.b, even though the latter retains enough functional information to activate *Ciinte.Mesp*.

In stark contrast to the rescue by Moocci.Tbx6-r.b, *Moocci.Mesp* reporter expression was never observed in *C. intestinalis* embryos upon overexpression of Moocci.Lhx3/4.b or Ciinte.Lhx3/4 in the posterior pole (*Figure 5L–N*). This indicates that a functional modification extending beyond *cis/trans* co-evolution accounts for the lack of *Moocci.Mesp* reporter activity in *C. intestinalis* embryos.

## Moocci.Lhx3/4.b overexpression is not sufficient to activate ectopic *Moocci.Mesp* in *M. occidentalis* embryos

The heterologous experiments in *C. intestinalis* suggested that *Moocci.Mesp* regulation could be independent of Moocci.Lhx3/4.b. To test this hypothesis, we recapitulated the experiments in *M. occidentalis* embryos. We isolated *cis*-regulatory sequences from the *Moocci.Tbx6-r.b* gene (*Figure 4—figure supplement 2*) and used this driver to overexpress *Lhx3/4* coding sequences in the posterior pole of *M. occidentalis* embryos. All Lhx3/4 proteins tested failed to induce ectopic *Moocci.Mesp* activation (*Figure 5O–R*). These data suggest that *Moocci.Mesp* is not responsive to Moocci.Lhx3/4.b, even in permissive *Moocci.Tbx6-r.b*-expressing cells. This differs markedly from the results of equivalent experiments in *C. intestinalis*, indicating major changes to the *cis*-regulatory logic governing identical *Mesp* spatiotemporal expression patterns in *Ciona* and *Molgula*.

## Conserved requirement for MAPK activation in specification of migratory TVCs

Given the *cis*-regulatory re-wiring of *Mesp* regulation, we extended our analysis to search for other instances of acute DSD between *Molgula* and *Ciona*. We shifted our focus to the next major cell fate decision in the B7.5 lineage, namely the fate choice between TVCs and ATMs. In *C. intestinalis,*

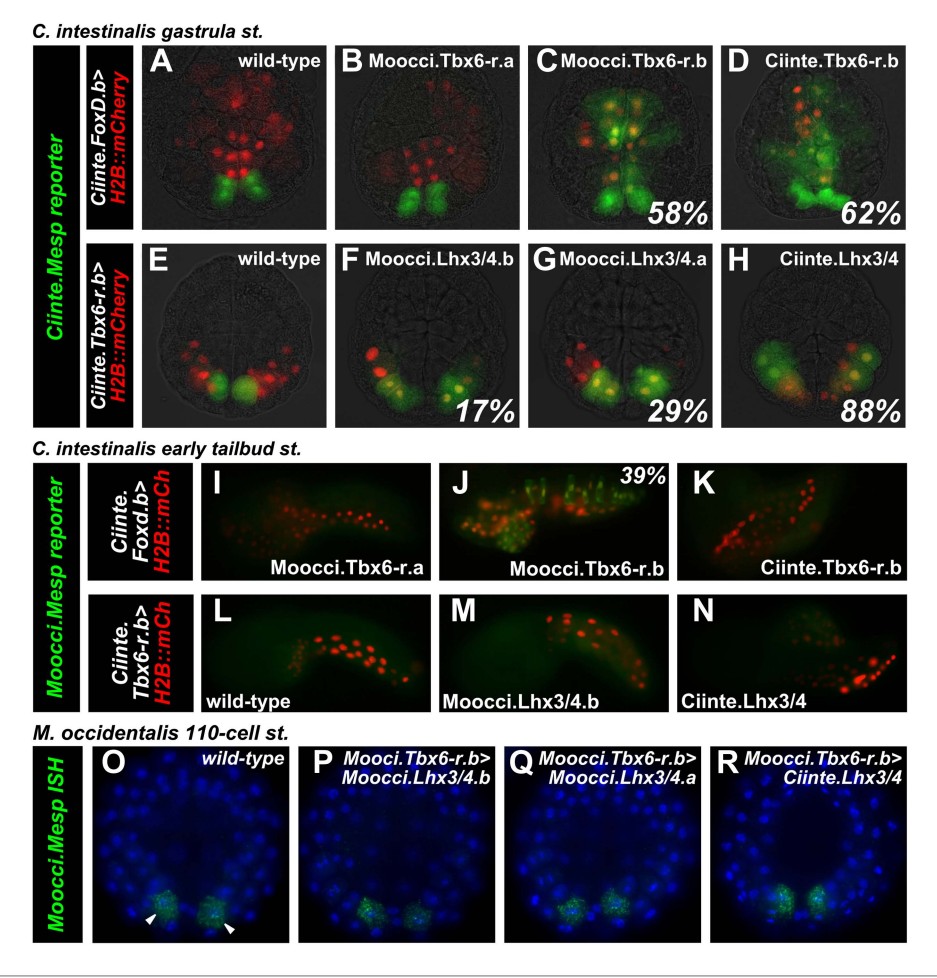

**Figure 5**. Divergence of *Mesp* regulation due to changes in *cis* and *trans*. (**A**) Wild-type *C. intestinalis* gastrula embryo showing expression of *Foxd.b* reporter (red, ***Shi and Levine, 2008***) in vegetal pole cells and *Ciinte.Mesp* reporter (green) in the B7.5 cells. (**B**) Electroporation of *Ciinte.Foxd.b>Moocci.Tbx6-r.a* has no effect on *Ciinte.Mesp* reporter expression. (**C**) Electroporation of *Ciinte.Foxd.b>Moocci.Tbx6-r.b* results in ectopic *Ciinte.Mesp* reporter expression in the vegetal pole in 58% of embryos (n = 100). (**D**) This is indistinguishable from the effect of *Ciinte.Foxd.b>Ciinte.Tbx6-r.b*, which results in ectopic *Ciinte.Mesp* reporter activation in 62% of embryos (n = 100). (**E**) Wild-type *C. intestinalis* gastrula embryo showing expression of *Tbx6-r.b* reporter (***Christiaen et al., 2009a***, red) in all B-line cells and *Ciinte.Mesp* reporter (green) in the B7.5 cells. (**F**) Electroporation of *Ciinte.Tbx6-r.b>Moocci.Lhx3/4.b* results in ectopic *Ciinte.Mesp* reporter expression in other B-line cells in 17% of embryos (n = 100) (**G**) *Ciinte.Tbx6-r.b>Moocci.Lhx3/4.a* results in ectopic *Ciinte.Mesp* reporter expression in 29% of embryos (n = 100). (**H**) *Ciinte.Tbx6-r.b>Ciinte.Lhx3/4* induces ectopic *Ciinte.Mesp* reporter expression in 88% of embryos (n = 100). (**I**) *C. intestinalis* tailbud embryo showing no expression of *Moocci.Mesp* reporter, as expected. (**J**) *Moocci.Mesp* reporter expression is not rescued by electroporation of *Ciinte.Foxd.b>Moocci.Tbx6-r.a* but (**K**) electroporation of *Ciinte.Foxd.b>Moocci.Tbx6-r.b* is sufficient to activate *Moocci.Mesp* reporter in 39% of embryos (n = 100). (**K**) *Moocci.Mesp* reporter is not transactivated upon electroporation of *Foxd.b>Ciinte.Tbx6-r.b*. (**L**) Wild-type *C. intestinalis* tailbud embryo, showing no expression of electroporated *Moocci.Mesp* reporter. (**M**) Electroporation of *Ciinte.Tbx6-r.b>Moocci.Lhx3/4.b* or (**N**) *Ciinte.Tbx6-r.b>Ciinte.Lhx3/4* is not sufficient to activate co-electroporated *Moocci.Mesp* reporter expression. (**O–R**) ISH showing *Moocci.Mesp* expression (green, arrowheads) in 110-cell stage *M. occidentalis* embryos. Ectopic *Moocci.Mesp* was not observed upon overexpression of any Lhx3/4 orthologs in the posterior embryo using the *Moocci.Tbx6-r.b* driver (**P–R**). All gastrula stage embryos are viewed vegetally, with anterior to the top of the image. All tailbud embryos are viewed laterally, anterior to the left. *M. occidentalis* embryo nuclei were visualized by staining with DAPI (blue).
*Figure 5. Continued on next page*

*Figure 5. Continued*

The following figure supplement is available for figure 5:

**Figure supplement 1**. Lhx3/4 proteins from *M. occidentalis* are sufficient to activate ectopic expression of endogenous *Mesp* in *C. intestinalis*.

the FGF/MAPK pathway, mediated in part by MEK-dependent phosphorylation and activation of Ets.b, is both necessary and sufficient for TVC specification. Inhibition of the FGF/MAPK/Ets pathway converts TVCs to ATMs, inhibiting cell migration and expression of TVC markers (*Davidson et al., 2006*; *Christiaen et al., 2008*; *Woznica et al., 2012*). Conversely, constitutive activation of the pathway induces ectopic TVCs at the expense of ATMs.

We ought to test the requirement of the MAPK pathway for TVC induction in *M. occidentalis.* We treated embryos with the MEK small molecule inhibitor U0126 prior to the estimated time window of TVC specification (~5 hpf at 22°C). U0126-treated embryos displayed inhibited TVC migration relative to DMSO-treated control embryos (*Figure 6A–D*). Moreover, expression of *Moocci.Foxf* and *Moocci.Hand-r* in the TVCs was severely downregulated by U0126 treatment relative to controls (*Figure 6A–D*). These data suggest a conserved requirement for MAPK pathway activity in TVC induction and expression of important TVC regulators in *M. occidentalis.*

We also tested the potential involvement of Ets factors in mediating the MAPK-dependent activation of the TVC program. To do so, we expressed a constitutively active fusion of the Ciinte.Ets.b DNA-binding domain to a VP16 transactivation domain (Ciinte.Ets.b::VP16, *Davidson et al., 2006*), in *M. occidentalis* B7.5 cells using the *Moocci.Mesp* driver. Electroporation of *Moocci.Mesp>Ciinte.Ets.b::VP16* induced ectopic TVCs at the expense of ATMs, in 31% of transfected embryos (*Figure 6E,F*). In sum, our results are consistent with a conserved function for MAPK-activated Ets.b upstream of the cardiopharyngeal gene regulatory network in *M. occidentalis.*

## *Foxf* TVC enhancers are mutually unintelligible in cross-species assays

Given the conservation of the MAPK/Ets pathway in ascidian TVC specification, we reasoned that *Foxf cis*-regulatory sequences should be compatible in cross-species assays. We identified a sequence upstream of the *Moocci.Foxf* gene that is sufficient to drive TVC-specific expression in *M. occidentalis* (*Figure 6G*, *Figure 6—figure supplement 1*). However, this sequence is completely non-functional when tested in *C. intestinalis* (*Figure 6H*). Likewise, a previously identified *Ciinte.Foxf* TVC enhancer (*Figure 6J*; *Beh et al., 2007*) is incapable of driving TVC-specific expression in *M. occidentalis* embryos (*Figure 6I*).

Thus, to borrow a term from the field of linguistics, these enhancers are said to be 'mutually unintelligible', since the enhancer from *M. occidentalis* cannot be interpreted by embryos of *C. intestinalis*, and the enhancer from *C. intestinalis* cannot be interpreted by *M. occidentalis*. This mutual unintelligibility is due to differences between the enhancers and not the promoters, since we used the same *Moocci.Foxf* basal promoter in both constructs. Although the *Ciinte.Foxf* enhancer is able to interact with and potentiate transcription from *Moocci.Foxf* promoter, nucleotide sequence is poorly conserved around *Foxf* proximal promoter regions, even between *M. occidentalis* and *M. occulta/oculata* (*Figure 6—figure supplement 1*).

We asked whether the observed unintelligibility was indicative of changes in Ets.b protein properties between *C. intestinalis* and *M. occidentalis* embryo. At first glance, differences in DNA binding domain amino acid sequences between the Ets.b orthologs do not predict a change in binding site preference (*Donaldson et al., 1996*). Indeed, full-length Moocci.Ets.b and Moocci.Ets.b::VP16 were just as effective as their *C. intestinalis* counterparts to induce ectopic TVCs when expressed in the *C. intestinalis* B7.5 lineage (*Figure 6—figure supplement 2*). Moreover, *C. intestinalis* TVC induction was abolished by expression of the constitutive repressor form (Moocci.Ets.b::WRPW). However, neither full-length Moocci.Ets.b nor Moocci.Ets.b::VP16 was sufficient to induce activation of *Moocci.Foxf* reporter in *C. intestinalis* embryos (*Figure 6—figure supplement 3*). These surprising results suggest that, even though TVC induction and TVC-specific *Foxf* expression depends upon Ets.b-mediated MAPK activity in *M. occidentalis* embryos, there are profound species-specific differences in the regulatory mechanisms acting in parallel to MAPK/Ets.b upstream of *Foxf*, reflected in the mutual unintelligibility of the orthologous enhancers.

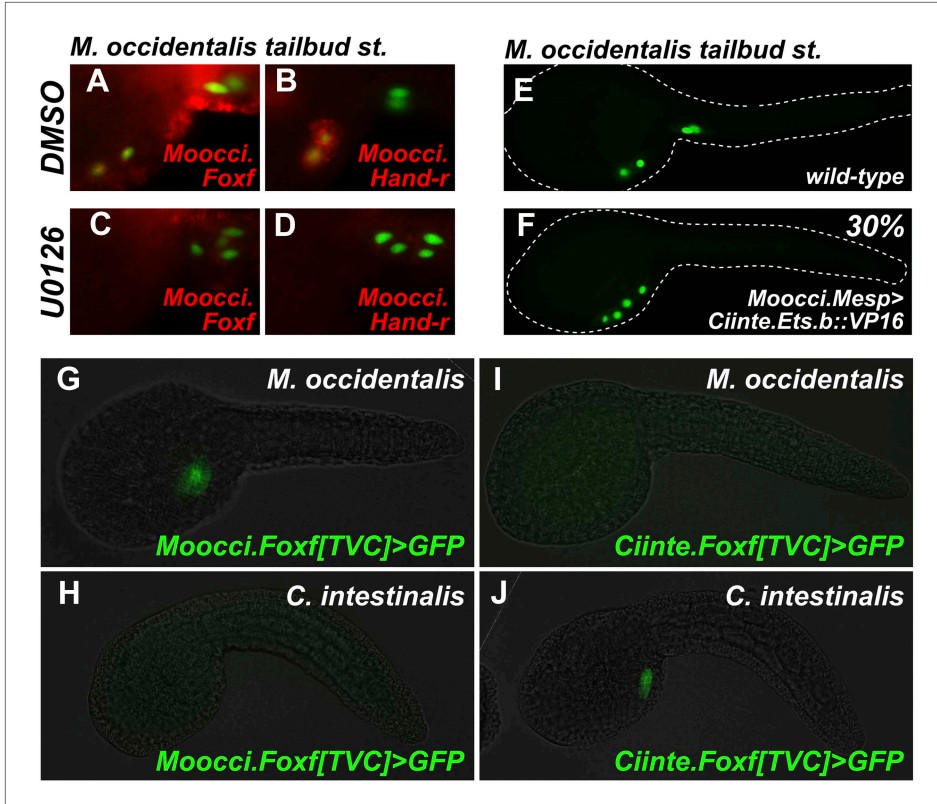

**Figure 6**. Mutual unintelligibility of orthologous *Foxf* enhancers in spite of shared requirement for the MAPK pathway in TVC fate specification. In situ hybridization (ISH, red) + β-galactosidase immunodetection (IHC, green) in embryos electoporated with *Moocci.Mesp>nls::lacZ* and treated with DMSO (control treatment), showing normal TVC induction in 85% of embryos (n = 49) and TVC-localized expression of (**A**) *Moocci.Foxf* and (**B**) *Moocci.Hand-r* in 96% (n = 25) and 100% (n = 23) of embryos, respectively. Treatment with MEK inhibitor U0126 inhibited TVC induction and migration in 66% of embryos (n = 128). Treatment with U0126 also abolished TVC-specific (**C**) *Moocci.Foxf* expression in 91% of embryos (n = 64) and (**D**) *Moocci.Hand-r* expression in 91% of embryos (n = 64). (**E**) Wild-type *M. occidentalis* tailbud embryo with B7.5 lineage labeled by electroporation of *Moocci.Mesp>H2B::GFP*. Note two TVCs and two ATMs. (**F**) TVC induction and migration was enhanced upon electroporation with *Moocci.Mesp>Ciinte. Ets.b::VP16,* which converted ATMs into TVC-like cells in 30% of electroporated embryos (n = 200). (**G**) A fragment from −1517 to −425 upstream of the *Moocci.Foxf* start codon (*Moocci.Foxf[TVC]*), fused to the minimal promoter (−230/+21) from *Moocci.Foxf* and GFP (*bpMoFF>GFP*), recapitulates expression in the TVCs of *M. occidentalis*. (**H**) This same exact construct is silent in *C. intestinalis* embryos. (**I**) Conversely, a published TVC enhancer from the *Ciinte.Foxf* gene (**Beh et al., 2007**) fused to *bpMoFF>GFP* is silent in *M. occidentalis* embryos. (**J**) This same construct is strongly expressed in the TVCs of *C. intestinalis*. All panels represent lateral views.

The following figure supplements are available for figure 6:

**Figure supplement 1**. Alignment of *Foxf* coding and non-coding sequences.

**Figure supplement 2**. Moocci.Ets.b and Ciinte.Ets.b are functionally very similar.

**Figure supplement 3**. Ets.b proteins are not sufficient to transactivate *Moocci.Foxf* reporter construct activation in *C. intestinalis* embryos.

## Extensive *cis*-regulatory unintelligibility between *C. intestinalis* and *M. occidentalis*

Since investigation into both *Mesp* and *Foxf* regulation revealed divergent *cis*-regulatory logic underlying identical gene expression patterns, we asked whether these examples constituted a general trend between *C. intestinalis* and *M. occidentalis*. Testing several other candidate enhancers, we identified

other examples of *cis*-regulatory unintelligibility. For instance, *Moocci.Tbx6-r.b* and *Mooci.Hand-r* upstream sequences recapitulated their endogenous activity in *M. occidentalis* embryos (***Figure 7B,E***), but these sequences were conspicuously silent in the B7.5 lineage in *C. intestinalis* embryos (***Figure 7C,F***). Notably, activity in other tissues seemed normal: *Moocci.Tbx6-r.b* reporter expression was robust and precise in *C. intestinalis* primary tail muscle cells (***Figure 7A,C***) and *Moocci.Hand-r* reporter recapitulated expression in the *C. intestinalis* A7.6 and anterior endoderm lineages (***Figure 7D,F***). In the case of *Hand-r,* this partial intelligibility is due to the modular organization of its *cis*-regulatory sequences, which in *C. intestinalis* is comprised of discrete enhancers controlling expression in each of the distinct domains (***Davidson and Levine, 2003***; ***Woznica et al., 2012***).

Curiously, we found that the *Ciinte.Hand-r* reporter can drive reporter gene expression in *M. occidentalis* TVCs (***Figure 7—figure supplement 1***). Thus, unlike the case of *Foxf*, there is an asymmetric intelligibility of *Hand-r* TVC enhancers between *M. occidentalis* and *C. intestinalis*. Moreover, a *M. oculata Hand-r* TVC enhancer is functional in *M. occidentalis* but not in *C. intestinalis* (***Figure 7—figure supplement 1***). Taken together, these data suggest that differences in enhancer logic may have

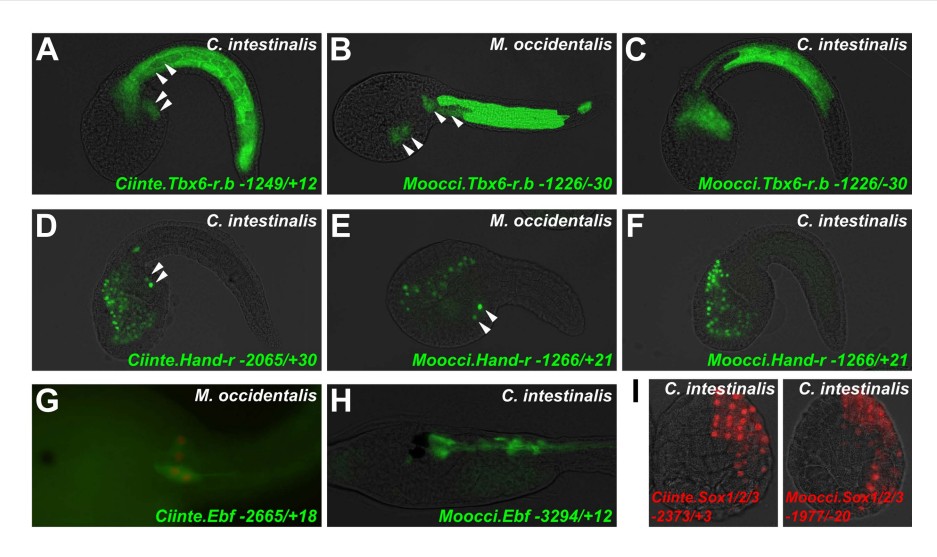

**Figure 7**. Cross-species reporter construct assays reveal multiple cases of *cis*-regulatory unintelligibility. (**A**) *C. intestinalis* embryo electroporated with *Ciinte.Tbx6-r.b>GFP* reporter construct (***Christiaen et al., 2009a***), which drives GFP expression in tail muscles and the B7.5 lineage cells (arrowheads), recapitulating endogenous *Ciinte.Tbx6-r.b* expression. (**B**) *M. occidentalis* embryo electroporated with *Moocci.Tbx6-r.b>GFP* reporter construct, which recapitulates expression in tail muscle cells including B7.5 lineage cells (arrowheads). (**C**) *C. intestinalis* embryo electroporated with *Moocci.Tbx6-r.b>GFP,* which drives expression in B-line tail muscle and mesenchyme cells but is excluded from the B7.5 lineage. (**D**) *C. intestinalis* embryo electroporated with *Ciinte.Hand-r>H2B::GFP* reporter (***Davidson and Levine, 2003***), which drives H2B::GFP expression in anterior endoderm, A7.6 lineage, and TVCs (arrowheads), recapitulating endogenous *Ciinte.Hand-r* expression. (**E**) *M. occidentalis* embryo electroporated with *Moccci.Hand-r>H2B::GFP* construct, which recapitulates the same expression pattern. (**F**) *C. intestinalis* embryo electroporated with *Moocci.Hand-r>H2B::GFP,* which drives expression in endoderm and A7.6 lineage, but not in B7.5. (**G**) *M. occidentalis* embryo electroporated with *Ciinte.Ebf* neuron-specific YFP (green) and H2B::mCherry (red) reporter constructs electroporated (***Stolfi and Levine, 2011***), which drive very weak expression in a limited subset of motor ganglion neurons (green and red). (**H**) *C. intestinalis* embryo electroporated with a *Moocci.Ebf>YFP* reporter, which drives robust reporter gene expression in several brain, motor ganglion, and nerve cord neurons. (**I**) *C. intestinalis* embryos electroporated with *Ciinte.Sox1/2/3* (left) and *Moocci.Sox1/2/3* (right) *H2B::mCherry* reporter constructs, both of which recapitulate *Sox1/2/3* expression in ectoderm. Panels **A–F** are lateral views of tailbud embryos, panels **G** is a dorsal view of a tailbud embryo, panel **H** is a dorso-lateral view of a hatched larva, and panel **I** shows vegetal views of mid-gastrula stage embryos. Anterior is to the right, except in Panel **I**, in which anterior is to the top.

The following figure supplement is available for figure 7:

**Figure supplement 1**. Asymmetric intelligibility between *Molgula* and *Ciona Hand-r* TVC enhancers.

accumulated over the course of the deep evolutionary history between *Molgula* and *Ciona* but not between *M. occidentalis* and *M. oculata,* and that some enhancers may have evolved asymmetrically in the two branches, retaining greater pan-ascidian 'intelligibility' in one or the other.

Finally, we found that acute DSD was not restricted to *cis*-regulatory sequences controlling gene expression in the B7.5 lineage. For instance, a *Ciinte.Ebf* driver that has been shown to recapitulate neuronal *Ebf* expression (*Stolfi and Levine, 2011*) was only very weakly active in a subset of *M. occidentalis* motor ganglion neurons (*Figure 7G*). However, the orthologous sequence from *M. occidentalis* recapitulated robust reporter gene expression in several *C. intestinalis Ebf+* neurons throughout the central nervous system (*Figure 7H*). In contrast, expression of a *Moocci.Sox1/2/3* reporter construct in the ectoderm of *C. intestinalis* embryos was found to be indistinguishable from that of the *C. intestinalis* ortholog (*Figure 7I*), suggesting that some *cis*-regulatory sequences may retain mutual intelligibility between *Molgula* and *Ciona* without sequence conservation, as is the case for various *Otx* enhancers between *C. intestinalis* and *H. roretzi* (*Oda-Ishii et al., 2005*). In sum, our preliminary cross-species enhancer activity screen suggests that DSD was rampant during ascidian evolution, resulting in mutual unintelligibility of *cis*-regulatory mechanisms underlying otherwise identical expression patterns of orthologous genes.

## Discussion

### *Molgula spp.* as model organisms for molecular developmental genetics

By sequencing the genomes of three *Molgula* species and adapting the electroporation protocol for routine transfection of *M. occidentalis* embryos, we have further expanded the reach of ascidian molecular genetics. We have demonstrated the usefulness of *M. occidentalis* as a species amenable to medium-throughput molecular perturbations and assays. While *Ciona/Halocynthia* comparisons have yielded fundamental insights into chordate development, the lack of a reliable electroporation protocol for transfection of *Halocynthia* eggs has hampered comparative studies on a larger scale. On the other hand, eggs from species such as *Phallusia mammilata* can be electroporated, but this species is much more closely related to *Ciona*. Thus, *Molgula* species may significantly contribute to understanding the extreme conservation of ascidian embryogenesis in spite of vast genetic differences and deep evolutionary histories.

We have shown that de novo sequencing and assembly of Molgulid genomes using next-generation technologies is relatively efficient and cost-effective, largely due to their compact nature. The sequence information obtained can be used to quickly generate a comprehensive set of molecular tools. We predict that a large number of ascidian genomes will be as compact and efficiently sequenced and assembled as these, unlocking the experimental potential of many ascidian species representing several different families (*Tsagkogeorga et al., 2009*).

### Similarities and differences between *Molgula* and *Ciona* cardiopharyngeal regulatory mechanisms

In our survey of B7.5 development, we observed that the cell division events and cell fate choices in the lineage are perfectly conserved between *Ciona* and *Molgula,* demonstrating the deep evolutionary origins of the cardiopharyngeal ontogenetic motif in tunicates. Regulatory gene expression patterns in the lineage are also nearly identical, the only potential difference being *Nk4,* which was not confidently detected in *M. occidentalis* TVCs.

NK4 proteins are intimately linked to cardiac development throughout bilateria (*Akazawa and Komuro, 2005*). Moreover, *Nk4* is expressed in *C. intestinalis* TVCs where it functions as a regulator of heart precursor fate (*Davidson and Levine, 2003*; *Wang et al., 2013*). However, there are distinct early and late phases of *Nk4* expression in *C. intestinalis* heart development. *Nk4* is expressed in the TVCs during the mid-tailbud stage, downregulated during the late tailbud and larval stages, and only re-activated in the differentiating heart of metamorphosing juveniles (*Davidson and Levine, 2003*). This points to distinct early (pre-metamorphic) and late (post-metamorphic) roles for *Nk4* in *C. intestinalis*.

In *C. intestinalis,* early *Nk4* expression does not appear to be sufficient for heart fate, as it is expressed in the TVCs prior to the segregation of heart and pharyngeal muscle fates (*Davidson and Levine, 2003*; *Beh et al., 2007*; *Christiaen et al., 2008*, *2010*; *Ragkousi et al., 2011*; *Wang et al., 2013*). Second, this early expression may not be strictly required for heart fate either, as the first heart

precursors (FHPs) do not adopt a pharyngeal muscle fate upon dominant-negative Nk4 overexpression like the second heart precursor (SHPs) do (*Wang et al., 2013*). Therefore, Nk4 acts as partially redundant mechanism to specify heart precursors in *C. intestinalis*, complementing parallel mechanisms of heart/pharyngeal muscle cell fate choice (e.g., activation of *Tbx1/10* in the STVCs and *Ebf* in the ASMFs).

It is possible that *Moocci.Nk4* is only activated during metamorphosis, an early role for it being dispensable in *M. occidentalis* TVCs. This might be the case if the complementary mechanism for pharyngeal muscle fate choice is more robust than in *Ciona*. In this way, the divergence of parallel regulatory mechanisms may have contributed to the loss of a key regulator like *Nk4* from the early cardiopharyngeal mesoderm GRN of *Molgula*. A much more detailed investigation will be required to determine a possible role for Nk4 in the *Molgula* heart development.

## Species-specific differences in B7.5 lineage morphogenetic events

*M. occidentalis* and *C. intestinalis* also display differences in certain morphogenetic behaviors of B7.5 derivatives (summarized in *Figure 3*, *Figure 3—figure supplement 3*). First, *M. occidentalis* TVCs migrate more laterally and do not meet at the ventral midline, resulting in a temporary condition of *cardia bifida*. A similar condition can be induced in *C. intestinalis* embryos by perturbing endoderm development (*Ragkousi et al., 2011*). Since *M. occidentalis* and *C. intestinalis* tailbud embryos look quite different in their shape, it will be interesting to assess endoderm formation and its relationship with TVC migration trajectory in *Molgula.*

Secondly, the ASM precursors in *M. occidentalis* migrate away from the heart progenitors but do not encounter atrial siphon primordia to encircle in the dorsal part of the embryo. This process is uncoupled in *C. intestinalis Hox1* mutants, which fail to specify the ASPs. In these mutant larvae, the ASMPs still appear to migrate dorsally but do not form rings of muscle (*Sasakura et al., 2012*). It remains to be determined how the *M. occidentalis* ASMPs migrate to the general vicinity of the future atrial siphon opening and remain localized there until the later specification and differentiation of the siphon.

These two major morphogenetic differences prompted us to rethink our assumptions regarding the interdependence of certain processes in ascidian cardiopharyngeal development, as identified in *C. intestinalis.* For instance, one might assume that the convergence of *Ciona* TVCs at the ventral midline to be important for the asymmetric cell divisions and fate choices segregating heart from ASM precursors. Similarly, one might assume the atrial siphon primordia to be the source of a cue required for the dorsal migration of ASMPs. In *M. occidentalis,* these processes have been naturally uncoupled. These observations serve to emphasize that sampling diverse taxa provides insights into the modularity of specific developmental processes.

## Developmental system drift hidden beneath the surface of highly conserved ascidian embryos

From our initial survey of a handful of enhancers from *C. intestinalis, M. occidentalis*, and *M. oculata*, we encountered several instances of either mutual unintelligibility, or asymmetric intelligibility of enhancers. Qualitative cross-species assays of *M. occidentalis* and *C. intestinalis cis*-regulatory sequences indicated that specific orthologous pairs of enhancers may be mutually intelligible (e.g., *Sox1/2/3* ectodermal enhancer) while others are mutually unintelligible (e.g., *Foxf* TVC enhancer*)*. Some orthologous gene pairs appear to be regulated by a mix of intelligible and unintelligible tissue-specific enhancers (e.g., *Hand-r).* The observation that cross-species intelligibility due to DSD occurs in a tissue-specific manner is consistent with the widely documented modularity of *cis*-regulatory elements controlling complex gene expression (*Arnone and Davidson, 1997*; *Levine, 2010*).

These results add to the mounting evidence suggesting that acute and pervasive DSD may have occurred over the course of ascidian evolution, obfuscated by the identical cell lineages and highly conserved gene expression patterns of ascidian embryos (*Swalla, 2004*). The multiple examples of *cis*-regulatory unintelligibility we identified were rather unexpected given (a) the extremely conserved pattern of expression of orthologous genes from *Molgula* and *Ciona* (*Takada et al., 2002*) and (b) previous observations of mutual intelligibility of enhancers between *C. intestinalis* and *H. roretzi* (e.g., *Otx*), and between *C. intestinalis* and the more closely related *C. savignyi* (*Johnson et al., 2004*; *Brown et al., 2007*). Large-scale, quantitative cross-species assays and detailed GRN studies will illuminate factors that may contribute to conservation or divergence in regulatory mechanisms.

## Evolution of different activating inputs for *Mesp* expression in ascidians

*Cis*-regulatory unintelligibility can arise due to different underlying factors. We have presented evidence that, in the case of *Mesp*, this results in part from a change in an upstream activating input. In *C. intestinalis,* Lhx3/4 is an activator of *Mesp*, while in *M. occidentalis, Mesp* activation appears to be independent of Lhx3/4.b, despite the conserved expression pattern of this gene in vegetal pole cells. Regulation of *Mesp* in other ascidian species will need to be investigated in order to establish whether Lhx3/4-dependent activation of *Mesp* in ascidians is ancestral or derived. On the other hand, both *C. intestinalis* and *M. occidentalis* have retained a conserved role for Tbx6-r in regulating *Mesp*, a regulatory connection that may have arisen in the last common ancestor of tunicates and vertebrates (*Yasuhiko et al., 2006*; *Davidson et al., 2005*). In hindsight, it appears the weak expression of *Ciinte. Mesp* reporter constructs observed in the B7.5 cells of *M. occidentalis* embryos is not indicative of GRN conservation. This expression instead appears to be an artifact attributed in part to the conserved, overlapping expression profiles of Moocci.Tbx6-r.b and Moocci.Lhx3/4.b, even if this overlap is not instructive for endogenous *Moocci.Mesp* activation.

We can only speculate what is the connection, if any, between the re-wiring of *Mesp* regulation and the *Molgula*-specific duplication and sub-functionalization of *Lhx3/4*. Interestingly, in *C. intestinalis* and *H. roretzi*, the early and late functions of Lhx3/4 are partitioned between two transcript variants of the same locus, not between different loci. This substitution of alternate isoforms with gene duplicates has been documented for a few genes in teleost fish (*Altschmied et al., 2002*; *Yu et al., 2003*). Although allowing for some degree of sub-functionalization of protein function, alternative transcription/splicing does not completely relax the constraints imposed by pleiotropy. The Lhx3/4 variants in *C. intestinalis* and *H. roretzi* still share the bulk of their amino acid sequence, differing only at the portion of the N-terminus encoded by the alternate first exons (*Christiaen et al., 2009a*; *Kobayashi et al., 2010*). In *Molgula* spp. on the other hand, the two paralogs are considerably diverged from one another throughout their coding sequences, even though the divergence does not appear to alter DNA sequence-binding specificity. Pleiotropy has been hypothesized to both promote and suppress evolution, more specifically DSD (*Hansen, 2003*). Investigating the full breadth of Lhx3/4 functions in multiple ascidian species will be required in order to determine the relationship between Lhx3/4 duplication and re-wiring of *Mesp* regulation.

## Multiple cases of cardiopharyngeal mesoderm enhancer unintelligibility point to differences in B7.5/TVC *trans* environments

The mutual unintelligibility of orthologous *Foxf* enhancers was unexpected, given that both *M. occidentalis* and *C. intestinalis* depend upon a shared MAPK/Ets-based switch for TVC fate induction. This discrepancy could be due to species-specific requirements for additional activating inputs, and/or silencing by species-specific repressors. Our observation that *cis-trans* complementation with Moocci.Ets.b::VP16 does not rescue the activity of the *Moocci.Foxf* enhancer in *C. intestinalis* embryos suggests that the latter possibility may be the case. It is thought that co-evolution between an enhancer and its *trans* environment is critical in order for it to faithfully direct a conserved pattern of gene expression, even if the overarching GRN is broadly conserved (*Landry et al., 2005*). Therefore, the incompatibility of B7.5 lineage-specific enhancers between *M. occidentalis* and *C. intestinalis* may reflect key differences in the *trans*-regulatory milieus of this lineage in the two species. Future comparative expression profiling of *M. occidentalis* and *C. intestinalis* B7.5 lineage cells will shed light on the nature of these differences.

## Concluding remarks

Even though the word 'drift' implies a greater role for chance over natural selection, adaptive processes are very likely to play a role in DSD. True and Haag hypothesized as much in their seminal introduction to the concept of DSD, predicting that *cis/trans* co-evolution and other compensatory changes should result in enhancers resembling species-specific 'Rube Goldberg machines' that give superficially similar outputs but are functionally non-interchangeable (*True and Haag, 2001*). Models incorporating selection, pleiotropy, and compensation (SPC) have refined this idea by ascribing 'drift' in one biological process to adaptation in another (*Landry et al., 2005*; *Johnson and Porter, 2007*; *Pavlicev and Wagner, 2012*; *Martinez et al., 2014*). For instance, if the same pleiotropic gene is involved in both processes, any mutations in that gene that are adaptive for the process under selection will require some compensatory changes simply to maintain the same output of the other process.

This may occur with or without deleterious intermediate states, with the latter occurring through a process termed 'pseudocompensation' (*Haag, 2007*).

We speculate that a high frequency of compensatory changes, required for the rapidly evolving ascidians to accommodate the constraints imposed by their invariant embryonic cell lineages and highly compact genomes, has given rise to a preponderance of cross-species *cis*-regulatory unintelligibility, following the DSD/SPC model. This perfect storm of intrinsic factors may be the key to explaining the dichotomy observed between highly conserved embryos and divergent *cis*-regulatory structure/function in ascidians.

# Materials and methods

## Genomic DNA library preparation and sequencing

Genomic DNA was phenol/chloroform extracted from dissected gonads of *Molgula occulta* (Kupffer) and *Molgula oculata* (Forbes) adults from Roscoff, France, and a *Molgula occidentalis* (Traustedt) adult from Panacea, Florida, USA (Gulf Specimen Marine Lab). Genomic DNA was sheared using an M220 Focused-ultrasonicator (Covaris, Woburn, MA). Sequencing libraries were prepared using KAPA HiFi Library Preparation Kit (KAPA Biosystems, Wilmington, MA) indexed with DNA barcoded adapters (BioO, Austin, TX). Size selection was performed using Agencourt (Beckman–Coulter, Brea, CA) AMPure XP purification beads (300–400 bp fragments), or Sage Science (Beverly, MA) Pippin Prep (650–750 bp and 875–975 bp fragments). For *M. occulta* and *M. occidentalis* libraries, 6 PCR cycles were used. For *M. oculata* libraries, 8 cycles were used for the 300–400 bp library, and 10 cycles were used for the 650–750 and 875–975 bp libraries. Libraries of different species but same insert size ranges were multiplexed for sequencing in three 100 × 100 PE lanes on a HiSeq 2000 sequencing system (Illumina, San Diego, CA) at the Genomics Sequencing Core Facility, Center for Genomics and Systems Biology at New York University (New York, NY). Thus, each lane was dedicated to a mix of species, specifically barcoded libraries of a given insert size range. Raw sequencing reads were deposited as a BioProject at NCBI under the ID# PRJNA253689.

## Genome sequence assembly

All genomes were assembled on Michigan State University High Performance Computing Cluster (http://contact.icer.msu.edu). Prior to assembly, read quality was examined using FastQC v0.10.1 (*Babraham Bioinformatics, 2014*). Reads were then quality trimmed on both the 5′ and 3′ end using seqtk trimfq (https://github.com/lh3/seqtk) which uses Phred algorithm to determine the quality of a given base pair. Seqtk trimfq only trims bases, so no reads were discarded. Each library per species was then abundance filtered using 3-pass digital normalization to remove repetitive and erroneous reads (*Brown et al., 2012*; *Schwarz et al., 2013*; *Howe et al., 2014*). Genome assembly was done using Velvet v1.2.08 (*Zerbino and Birney, 2008*) with k-mer overlap length ('k') ranging from 19 to 69 and scaffolding was done by Velvet, by default. Velvet does not produce separate files for contigs and scaffolds; because Velvet scaffolded conservatively, contigs dominated the assemblies so we refer to both contigs and scaffolds as contigs. CEGMA scores were then computed to evaluate genome completeness (*Parra et al., 2007*).

The latest versions of three species' genome assemblies have been deposited on the ANISEED (Ascidian Network for In Situ Expression and Embryological Data) database for browsing and BLAST searching at http://www.aniseed.cnrs.fr/ (*Tassy et al., 2010*). Scripts for genome assembly and CEGMA analysis can be found in the following github repository: https://github.com/elijahlowe/molgula_genome_assemblies.git.

## Molecular cloning of *Molgula* sequences

Putative orthologs of *Ciona intestinalis* (Linnaeus) and *Halocynthia roretzi* (Drasche) protein-coding genes were initially identified by TBLASTN. Identified sequences were aligned to each other to support orthology relationships within *Molgula*, and then queried by BLASTP against the NCBI non-redundant protein sequence database to further support orthology relationships to known tunicate and vertebrate proteins. cDNAs were cloned by RT-PCR, or by 5′ and/or 3′ RACE (SMARTer RACE kit, Clontech, Mountain View, CA). The template cDNA libraries were prepared from total RNA extracted from embryos of various stages. Upstream regulatory sequences were cloned by PCR from genomic DNA. For non-RACE PCRs, we used Phusion high-fidelity polymerase (New England Biolabs, Ipswich, MA). Refer to *Supplementary file 1* for detailed sequence information. Genes are named according to the proposed unified nomenclature system for tunicate genetic elements (*Stolfi et al., 2014*).

Refer to *Supplementary file 2* for table with previously used gene names and the corresponding new proposed gene symbols and names.

## Electroporation of plasmid DNA into embryos

*M. occidentalis* gravid adults were obtained from Gulf Specimen Marine Lab between March and December. Gravid adults were rare between the winter months from December to March. Animals were kept in a small aquarium with circulating artificial sea water at 24°C until needed. All subsequent manipulations using eggs and embryos were performed between 20°C and 24°C, though 24°C appeared to be the optimal temperature for both dechorionation and embryonic development.

Individual gonads were dissected in filtered TAPS-buffered artificial sea water (T-ASW) to release eggs and sperm. Eggs were allowed to undergo germinal vesicle breakdown for 10 min and then pooled along with sperm to fertilize for 5 min. Eggs were rinsed thoroughly and dechorionated in solution (168 µl 10N NaOH, 1.2 ml 2.5% pronase E, 0.3 g sodium thioglycolate in 40 ml T-ASW) with constant pipetting for 8–10 min (modified after *Christiaen et al., 2009c*). Viewing under a high-powered dissecting microscope was often necessary to verify removal of the thin, translucent chorion that becomes wrapped tightly around the egg upon immersion in dechorionation solution. After confirming chorion removal, eggs were quickly rinsed by the passage through multiple dishes of T-ASW.

Electroporation conditions and procedures were identical to those previously described for *C. intestinalis* (*Christiaen et al., 2009b*). Unless otherwise specifically noted, all fluorescent reporter genes were fused at the N-terminus to unc-76 tags (*Dynes and Ngai, 1998*), to ensure distribution throughout the whole cytosol, electroporated typically at ~50 µg to 90 µg per 700 µl electroporation solution, while nuclei were visualized with histone2B::fluorescent protein constructs electroporated at ~5 µg to 25 µg per electroporation unless otherwise specifically noted. Perturbation constructs were electroporated at ~35 µg to 100 µg per electroporation. To image juveniles, larvae were allowed to settle and metamorphose in plastic Petri dishes or on glass coverslips in Petri dishes for several days with frequent changes of T-ASW.

Fertilization of *M. occulta* eggs was performed as previously described (*Swalla and Jeffery, 1990*). Embryos were dechorionated and electroporated as described for *M. occidentalis* embryos, using a custom-built electroporation machine (*Zeller et al., 2006*). *C. intestinalis* (Type A '*robusta*') adults were collected in San Diego, CA (M-Rep).

## MEK1/2 inhibitor U0126 treatment

Embryos were treated in 10 µM U0126 (Cell Signaling Technology, Danvers, MA) in T-ASW at least 30 min prior to the targeted MAPK signaling event.

## In situ hybridization

Fluorescent in situ hybridization assays were performed as previously described, with modifications (*Beh et al., 2007*; *Ikuta and Saiga, 2007*; *Christiaen et al., 2009d*). Embryos were fixed in MEM-3.2% to 4% paraformaldehyde buffers for at least 2 hr. Pre-hybridization proteinase K concentrations ranged from 0.25 µg/ml (110-cell stage) to 1 µg/ml (tailbud) and 5 µg/ml (larvae) for *Molgula spp.* embryos. The *Ciinte. Ebf* probe was synthesized from template plasmid from the *C. intestinalis* Gene Collection Release 1 (*Satou et al., 2002*). See *Supplementary file 1* for more detailed probe template sequence information.

## Imaging

Embryos were counterstained with DAPI and/or Phalloidin conjugates (LifeTechnologies/Thermo Fisher Scientific, Waltham, MA). Images were taken using a combination of different Leica Microsystems (Wetzlar, Germany) microscopes: TCS SP8 X confocal microscope, TCS SP5 confocal microscope, and DM2500 epifluorescence microscope.

## Acknowledgements

The authors are greatly indebted to Paul Scheid, members of the NYU CGSB Genomics Sequencing Core facility, Rodoniki Athanasiadou and the Gresham Lab for help in planning and performing library preparation and sequencing. We thank ASSEMBLE MARINE for funding the MoEvoDevo project in 2013. We thank the Station Biologique faculty and staff, especially Stéphane Hourdez, Sophie Booker, and Xavier Bailly for their help in carrying out our studies at Roscoff. We would also like to thank Nadine Peyriéras and her research group for sharing of reagents, equipment, and expertise for *M. occulta* and *M. oculata* imaging. We are grateful to Wei Wang for subcloning the full-length *Ciinte.Ets.b* sequence. Work in

the laboratory of LC is supported by grants from the National Institute of General Medical Sciences (R01GM096032), the American Heart Association (10SDG4310061), and by the New York Cardiac Center and New York University College of Arts and Sciences. AS is supported by the National Science Foundation Postdoctoral Research Fellowship in Biology (under grant NSF-1161835). EL, BJS, and CTB are supported by the National Science Foundation under Cooperative Agreement No. DBI-0939454. Any opinions, findings, and conclusions or recommendations expressed in this material are those of the author(s) and do not necessarily reflect the views of the National Science Foundation. This project was supported by Agriculture and Food Research Initiative Competitive Grant 2010-65205-20361 to CTB.

## Additional information

### Funding

| Funder | Grant reference number | Author |
|---|---|---|
| National Institute of General Medical Sciences | R01GM096032 | Lionel Christiaen |
| National Science Foundation | NSF-1161835 | Alberto Stolfi |
| Agriculture and Food Research Initiative | 2010-65205-20361 | C Titus Brown |
| ASSEMBLE MARINE | | Billie J Swalla |
| American Heart Association | 10SDG4310061 | Lionel Christiaen |
| New York Cardiac Center | | Lionel Christiaen |
| National Science Foundation | DBI-0939454 | Elijah K Lowe, C Titus Brown, Billie J Swalla |
| New York University College of Arts and Science | | Lionel Christiaen |

The funders had no role in study design, data collection and interpretation, or the decision to submit the work for publication.

### Author contributions

AS, EKL, BJS, Conception and design, Acquisition of data, Analysis and interpretation of data, Drafting or revising the article, Contributed unpublished essential data or reagents; CR, Conception and design, Acquisition of data, Contributed unpublished essential data or reagents; FR, Conception and design, Contributed unpublished essential data or reagents; CTB, Conception and design, Analysis and interpretation of data, Drafting or revising the article, Contributed unpublished essential data or reagents; LC, Conception and design, Acquisition of data, Analysis and interpretation of data, Drafting or revising the article

## Additional files

### Supplementary files

• Supplementary file 1. DNA sequences of probes, enhancers, promoters, protein-coding cDNAs, in situ hybridization probe templates, primers, etc used in this study.

• Supplementary file 2. Table of newly proposed tunicate gene names and symbols (*Stolfi et al., 2014*), and their aliases and synonyms.

### Major dataset

The following dataset was generated:

| Author(s) | Year | Dataset title | Dataset ID and/or URL | Database, license, and accessibility information |
|---|---|---|---|---|
| Stolfi A, Lowe EK, Racioppi C, Ristoratore F, Brown CT, Swalla BJ, Christiaen LC | 2014 | Molgula occidentalis, M. occulta, and M. oculata genome samples | http://www.ncbi.nlm.nih.gov/bioproject/PRJNA253689 | BioProject - NCBI. |

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
