## [Decision Letter]

Thank you for sending your work entitled “Identical cell lineages and gene expression patterns conceal developmental system drift in ascidian evolution” for consideration at *eLife.* Your article has been favorably evaluated by Diethard Tautz (Senior editor), a Reviewing editor, and 3 reviewers, one of whom, Michael Eisen, has agreed to reveal their identity.

The Reviewing editor and the other reviewers discussed their comments before we reached this decision, and the Reviewing editor has assembled the following comments to help you prepare a revised submission.

The paper represents an important contribution to the field of evolutionary biology and genome regulation. The experimental work is well executed. Before publication, the following points should be addressed by the authors:

1) As the authors outline in the Introduction, the lack of non-coding sequence conservation between distantly related ascidian species is established. Accordingly, while the authors reinforce and extend this, the extent to which their findings of divergent regulatory mechanisms are “unexpected” might be toned down.

2) The authors observe that Nk4 seems to be absent from the Molgula B7.5 lineage. The authors should discuss further how loss of key regulators could boost divergence in regulatory mechanisms. Don't the largely conserved gene expression profiles suggest that there may not be major changes in the trans-regulatory milieu as proposed in the Discussion?

3) The authors include Molgula occulta in their study. Can they draw more conclusions about this species? What happens to the equivalent of ATM cells in this tail-less species?

4) The differences in initial heart and AS formation between Ciona (single heart and bilateral siphon primordia) and Molgula (bifid heart and single AS primordia) are intriguing. Is it possible that this reflects a limited initial distribution of the B7.5 lineage around the circumference of the ascidian embryo?

5) Is the expression of regulators of the B7.5 lineage conserved at other sites of expression? For example, the authors mention that Foxf and Gata4/5/6 expression in surrounding tissue obscures TVC cells in Mogula. Is this also the case in Ciona?

6) Can the authors comment on the extent to which non-coding sequence divergence applies to the proximal promoters of ascidian genes with conserved expression patterns? Are these elements more conserved than enhancers?

7) More speculatively, can the authors make any conclusions about synteny from their genome sequence data? It would be interesting to know if the turnover of regulatory mechanisms is associated with as breakdown of synteny, as conserved gene order has been proposed to result from conservation of enhancer activity embedded in neighbouring genes.

8) It would help to have summary gene network schema highlighting similarities and differences between ascidian genera.

[Editors’ note: further minor revisions were requested prior to acceptance.]

There is one minor issue that we would like you to consider addressing prior to publication: concerning the use of “identical” in the title and abstract, presumably it implies that as far as the authors compared, the different species the lineages and expression patterns were the same (but Nk4 expression differences noted...). “Similar” may be more accurate and the reviewers and editor prefer that you substitute this term in the title, abstract, and where relevant in the text.

---

## [Author Response]

The most visible proposed change has been in the title from “Identical cell lineages and gene expression patterns conceal developmental system drift in ascidian evolution” to “Identical cardiopharyngeal lineages and expression patterns conceal developmental system drift in ascidian evolution”. We decided to change it in order to convey the notion that our study was nearly entirely focused on the cardiopharyngeal lineage. The original title had contained this key lineage information, but this was abandoned in haste at the moment of submission due to title size restrictions. We have now found a way to fit this back into the title. We hope you accept our reasoning for this title change.

*1) As the authors outline in the Introduction, the lack of non-coding sequence conservation between distantly related ascidian species is established. Accordingly, while the authors reinforce and extend this, the extent to which their findings of divergent regulatory mechanisms are “unexpected” might be toned down*.

We generally agree with the reviewers’ point and have altered the wording in the text in order to downplay our prior assumptions, which were proven to be misguided. However, we have also attempted to more clearly explain in the Introduction the distinctions between the qualitatively different manifestations of cryptic regulatory divergence described in the literature which have been broadly classified under the term developmental system drift (DSD).

At face value, the finding that divergent non-coding sequences underlie divergent regulatory mechanisms does not seem unexpected. However, we and other ascidian researchers assumed that *cis-*regulatory mechanisms must be highly conserved throughout the ascidians, in order to explain the remarkable conservation of invariant cell lineages and gene expression patterns between the very distantly related model species *C. intestinalis* and *H. roretzi.* Furthermore, the only in-depth comparative analysis of *cis-*regulatory sequence function between these two species revealed strong conservation of enhancer logic *in spite* of radically divergent sequences (71), mirroring the findings of studies in other groups of animals (59; 35; 79; 62; 75).

Examples of divergent regulatory mechanisms between *Ciona* and *Halocynthia* in the literature have been mostly limited to differences in extracellular signaling cues, not *cis*-regulatory logic (22; 1; 107; 40; 39; 102). Thus, when we began the present study, we were biased to expect no significant divergence in *cis-*regulatory mechanisms between *Molgula* and *Ciona,* especially not in those controlling the highly conserved expression patterns of genes such as *Mesp, Foxf,* and *Hand-related.* We were obviously wrong in our expectations, but we believe these were somewhat justified at the onset of our investigation.

*2) The authors observe that Nk4 seems to be absent from the Molgula B7.5 lineage*. *The authors should discuss further how loss of key regulators could boost divergence in regulatory mechanisms. Don't the largely conserved gene expression profiles suggest that there may not be major changes in the trans-regulatory milieu as proposed in the Discussion?*

As we have mentioned in the text, we cannot definitively conclude that *Nk4* is absent from the Molgula B7.5 lineage. We have created a new section of the Discussion dedicated to outlining more clearly both the caveats of our *Nk4* expression analysis and the hypothetical scenarios involving possible loss of (early) *Nk4* expression in *M. occidentalis* TVCs.

The reviewers suggest that the loss of key regulators could boost divergence in regulatory mechanisms. We generally agree with this statement. However, the converse is just as likely. For instance, it is possible that in *Molgula,* there is no need for the partially redundant function of Nk4 to promote SHP fate, because the complementary mechanism for pharyngeal muscle fate choice may be more robust than in *Ciona.* In this scenario, the divergence of a parallel (or upstream) regulatory mechanism may have prompted the loss of a key regulator. We have explicitly added these points in the Discussion.

*3) The authors include Molgula occulta in their study*. *Can they draw more conclusions about this species? What happens to the equivalent of ATM cells in this tail-less species?*

As far as we can tell, the primary larval muscle cells (including the ATMs) of *M. occulta* are specified but do not differentiate, partly because *M. occulta* have lost certain larval muscle structural genes (52). There is no evidence that there are any other fundamental differences in B7.5 lineage development between *M. occidentalis* and *M. occulta*.

We have included additional supplemental figures (Figure 2—figure supplement 3, panels G-I; Figure 2—figure supplement 4) showing *in situ* hybridization assays in *M. occulta* and *M. occidentalis* tailbud-stage embryos for *Aldh1a* (also known as *Raldh2*), which encodes the rate-limiting enzyme for retinoic acid biosynthesis*.* In *C. intestinalis, Aldh1a.a* is expressed in the TVCs and anterior primary muscle cells (including ATMs) and establishes a retinoic acid signaling gradient that plays important roles in patterning, fate specification, and differentiation in different tissues and cell lineages of the embryo (68).

From our data, we can observe that *Aldh1a* expression is conserved in *M. occidentalis* (Figure 2—figure supplement 4). Furthermore, it appears that *M. occulta* embryos have vestigial ATMs that remain in the anterior portion of the abortive tail bud and express *Aldh1a* (Figure 2—figure supplement 3)*.* This suggests that, although the ATM cells do not differentiate, they may still retain a function as an important signaling center for the embryo. We have updated the manuscript to include a brief description and discussion of these new data.

*4) The differences in initial heart and AS formation between Ciona (single heart and bilateral siphon primordia) and Molgula (bifid heart and single AS primordia) are intriguing*. *Is it possible that this reflects a limited initial distribution of the B7.5 lineage around the circumference of the ascidian embryo?*

Since the B7.5 lineage, like most other cell lineages of the ascidian embryo is invariant and always gives rise to a fixed, stereotyped number of cells that is conserved between *Ciona* and *Molgula,* the reviewers are correct in assuming this population of cells is limiting. However, we do not believe there is any connection between the arrangement of the heart precursors and the single- or dual-primordium mode of atrial siphon development. It appears that once the atrial siphon muscle progenitors have separated from the heart precursors, their fates become largely independent from one another.

It has been well documented that stolidobranch ascidians form a single atrial siphon primordium ([30]; Grave 1944), but we have not encountered any bipartite heart primordia previously described in the literature. This probably reflects a *Molgula-*specific condition that evolved well after the divergence of the stolidobranchs and their transition to a single-primordium condition. We see how the readers may be tempted to draw a correlation between the number of heart and atrial siphon primordia, and have modified the text to reinforce the message that the bifid heart is likely a *Molgula-*specific trait, while the single atrial siphon primordium is a pan-stolidobranch character.

*5) Is the expression of regulators of the B7.5 lineage conserved at other sites of expression? For example, the authors mention that Foxf and Gata4/5/6 expression in surrounding tissue obscures TVC cells in Mogula*. *Is this also the case in Ciona?*

Yes, this is the case in *Ciona* (7)*. Foxf and Gata4/5/6* expression in these tissues is perfectly conserved between *Ciona* and *Molgula.* We have included a line in the text that refers to this conserved expression in other tissues.

*6) Can the authors comment on the extent to which non-coding sequence divergence applies to the proximal promoters of ascidian genes with conserved expression patterns? Are these elements more conserved than enhancers*?

Annotation of the *Molgula* genomes is still ongoing and thus we are not able to answer this question on a genome-wide scale, but for the few genes we have analyzed in depth, the promoters are no more conserved at the nucleotide sequence level than the enhancers, that is to say, not at all. In fact, even coding sequences are poorly conserved, and alignment is usually only possible between sequences representing the DNA-binding domain or other functional domains. One can see this clearly in the VISTA alignment plots that we have already included in the manuscript (*Mesp,*
Figure 1, and *Foxf,*
Figure 6—figure supplement 1)*.*

However, it is clear that the *Moocci.Foxf* proximal promoter is fully functional in *C. intestinalis* embryos. This was demonstrated by generating chimeras between the *Ciinte.Foxf* TVC enhancer and the *Moocci.Foxf* proximal promoter, which worked in *C. intestinalis* but not in *M. occidentalis* to drive TVC gene expression. However, the converse (*Ciinte.Foxf* promoter in *M. occidentalis*) has not been tested. We have added a line highlighting these observations in the text.

*7) More speculatively, can the authors make any conclusions about synteny from their genome sequence data? It would be interesting to know if the turnover of regulatory mechanisms is associated with as breakdown of synteny, as conserved gene order has been proposed to result from conservation of enhancer activity embedded in neighbouring genes*.

We would not be comfortable making any conclusions about conservation of synteny or co-linearity, given that our assemblies did not use any long-insert library sequence reads, nor have we mapped our assemblies to chromosomes. Therefore, we have not made any changes to the text regarding this topic.

That being said, in a comparison between the *C. intestinalis* and *C. savignyi* genomes (Hill et al. 2008) observed extensive retention of synteny (genes located on the same chromosome) but an equally extensive loss of co-linearity (genes arranged in the same order and direction on a given chromosome). These findings seem to argue against an association between breakdown of colinearity and regulatory turnover, as regulatory mechanisms are quite conserved between *C. intestinalis* and *C. savignyi* (45).

On the other hand, tunicate genomes are quite compact (Dehal et al. 2002; Denoeud et al. 2010) and their cis-regulatory organization appears relatively simple, with all developmentally-important enhancers reported falling within <10 kb immediately 5’ to the transcription start site or within the introns of the regulated gene in question. To our knowledge there have been no reports, in *Ciona*, of enhancers embedded in one gene controlling the transcription of a neighboring gene. If we had to speculate, we would say that the compact nature of tunicate genomes has allowed for a breakdown of colinearity, given the lack of constraints imposed by the long-range enhancers mentioned by the reviewers.

*8) It would help to have summary gene network schema highlighting similarities and differences between ascidian genera*.

Strictly speaking, we do not have the evidence to infer gene networks operating in the B7.5 lineage of *Molgula.* The only regulatory connections we have directly established are *Tbx6-r.b* → *Mesp* and MAPK signaling → *Foxf/Hand-r. A*dditionally, the B7.5 network of *Ciona* is not substantially better understood. Therefore, we feel it is misleading to draw gene network schema at this stage of our understanding.

Better understood are the similarities and differences in the cell division events, basic morphogenesis, and gene expression patterns of the lineage. Of these, only the basic morphogenesis shows striking enough differences to warrant a comparative schematic diagram, which has been drawn and included in the revised version of the manuscript as Figure 3—figure supplement 3.

[Editors’ note: further minor revisions were requested prior to acceptance.]

*There is one minor issue that we would like you to consider addressing prior to publication: concerning the use of “identical” in the title and abstract, presumably it implies that as far as the authors compared, the different species the lineages and expression patterns were the same (but Nk4 expression differences noted...). “Similar” may be more accurate and the reviewers and editor prefer that you substitute this term in the title, abstract, and where relevant in the text*.

We would like to propose the following modified title, which conveys the main findings perhaps more clearly and accurately:

“Divergent mechanisms regulate conserved cardiopharyngeal development and gene expression in distantly related ascidians”

Please note that where we wrote “identical” in the Abstract, referring to cell division and fate specification events, we showed that these are indeed identical between the two species.